# MCM5 UFMylation regulates replication origin firing and fork progression

Zheng Li [ID][1,2,8], Xingxuan Wu[1,3,8], Liu Liu[4], Shaohong Rao[1], Yanting Liao[1], Mengting Liu[1], Bin Peng[1], Qiongdan Zhang[5], Yisui Xia [ID][6], Yuanliang Zhai [ID][5,7], Shunichi Takeda [ID][1✉] & Xingzhi Xu [ID][1✉]

## Abstract

**Modification with UFM1 (UFMylation) is essential for cell proliferation, but its precise mechanism of action is unclear. Furthermore, the UFMylation pathway has been associated with microcephalic primordial dwarfism (MPD) disorders, and mutations causative for MPD are also identified in genes encoding components of the replicative DNA helicase complex, including the MCM hexamer. Here, we reveal that UFMylation regulates DNA replication, and that all MPD-associated mutations in UFMylation enzymes impair replication. Mechanistically, the UFM1 E3 ligase UFL1 catalyzes Lys583 UFMylation of MCM5, a critical component of the CMG replicative DNA helicase complex. Mutation of Lys583 blocking this UFMylation event destabilizes the helicase complex, delaying origin firing and slowing replication fork progression. We conclude that MCM5 UFMylation is essential for efficient origin firing and replication fork progression, both of which ensure accurate DNA replication, cell proliferation, and prevention of MPD disorders.**

**Keywords** UFMylation; UFL1; CMG Helicase; Origin Firing; DNA Replication
**Subject Categories** DNA Replication, Recombination & Repair; Molecular Biology of Disease; Post-translational Modifications & Proteolysis

## Introduction

Ubiquitin-fold modifier 1 (UFM1) is a ubiquitin-like (UBL) protein that post-translationally modifies proteins in a process known as UFMylation (Komatsu et al, 2024). This modification event requires the E1 ubiquitin-like modifier-activating enzyme 5 (UBA5), the E2 UFM1-conjugating enzyme 1 (UFC1), and the E3 UFM1-specific ligase 1 (UFL1). UFMylation is involved in multiple pathways and cellular processes, including endoplasmic reticulum homeostasis (Li et al, 2018; Ishimura et al, 2023; Mao et al, 2023; Makhlouf et al, 2024; DaRosa et al, 2024), tumor formation (Liu et al, 2020; Wang et al, 2023), and the DNA damage response (Qin et al, 2019; Wang et al, 2019; Qin et al, 2022). Indeed, studies in mice underscore the critical importance of UFMylation, as a homozygous deletion of UBA5 or UFL1 causes embryonic lethality (Zhang et al, 2015; Tatsumi et al, 2011), likely due to a proliferation defect in UBA5 or UFL1-depleted cells. Despite these findings, how UFMylation regulates cell proliferation is unknown.

During the early stages of DNA replication, the CDC45-MCM-GINS (CMG) holo-helicase complex is assembled at replication origins (Lewis et al, 2022; Li et al, 2023; Xia et al, 2023; Xu et al, 2023), where it determines the origin firing and progression of replication forks (Xiang et al, 2023; Costa and Diffley, 2022; Cvetkovic et al, 2023; Jones et al, 2023; Xia et al, 2023; Terui et al, 2024; Parker et al, 2017). First, two MCM hexamers bind to replication origins in a head-to-head manner. Then, CDC45 (cell division cycle 45) and the protein complex known as "GINS" associate with a subset of the loaded MCM double-hexamers during the late $G_1$ to S phase to form an active form replicative helicase CMG (Xiang et al, 2023; Costa and Diffley, 2022; Gambus et al, 2006; Costa et al, 2011; Eickhoff et al, 2019; Langston et al, 2023). The regulation of budding yeast CMG assembly and activation have been well-studied in the past 20 years (Zegerman and Diffley, 2007; Tanaka et al, 2007; Remus et al, 2009; Yeeles et al, 2015; Ticau et al, 2015; Douglas et al, 2018); however, it remains unclear in mammalian cells. Indeed, it has only recently been shown in higher eukaryotes that the DNA repair and maintenance protein, DONSON is an essential component for loading GINS onto MCM hexamers (Cvetkovic et al, 2023; Evrin et al, 2023; Kingsley et al, 2023; Xia et al, 2023; Lim et al, 2023; Hashimoto et al, 2023; Terui et al, 2024), maintaining CMG helicase function during replication progression, and preventing microcephalic primordial dwarfism (MPD) disorders (Reynolds et al, 2017; Zhang et al, 2020).

Germline mutations in genes encoding components of the replication machinery have been associated with MPD disorders. Meier-Gorlin Syndrome exemplifies an MPD disorder arising from

[1]Guangdong Key Laboratory for Genome Stability & Disease Prevention and Carson International Cancer Center, Marshall Laboratory of Biomedical Engineering, Shenzhen University Medical School, Shenzhen University, Shenzhen, China. [2]Shenzhen University General Hospital-Dehua Hospital Joint Research Center on Precision Medicine (sgh-dhhCPM), Dehua Hospital, Dehua, China. [3]Guangdong Key Laboratory for Biomedical Measurements and Ultrasound Imaging, National-Regional Key Technology Engineering Laboratory for Medical Ultrasound, School of Biomedical Engineering, Shenzhen University Medical School, Shenzhen, China. [4]College of Life Sciences, Capital Normal University, Beijing, China. [5]School of Biological Sciences, The University of Hong Kong, Hong Kong, China. [6]South China Hospital, Guangdong Key Laboratory for Genome Stability & Disease Prevention, Shenzhen University Medical School, Shenzhen, China. [7]Division of Life Science, The Hong Kong University of Science & Technology, Hong Kong, China. [8]These authors contributed equally: Zheng Li, Xingxuan Wu. ✉E-mail: stakeda@szu.edu.cn; Xingzhi.Xu@szu.edu.cn

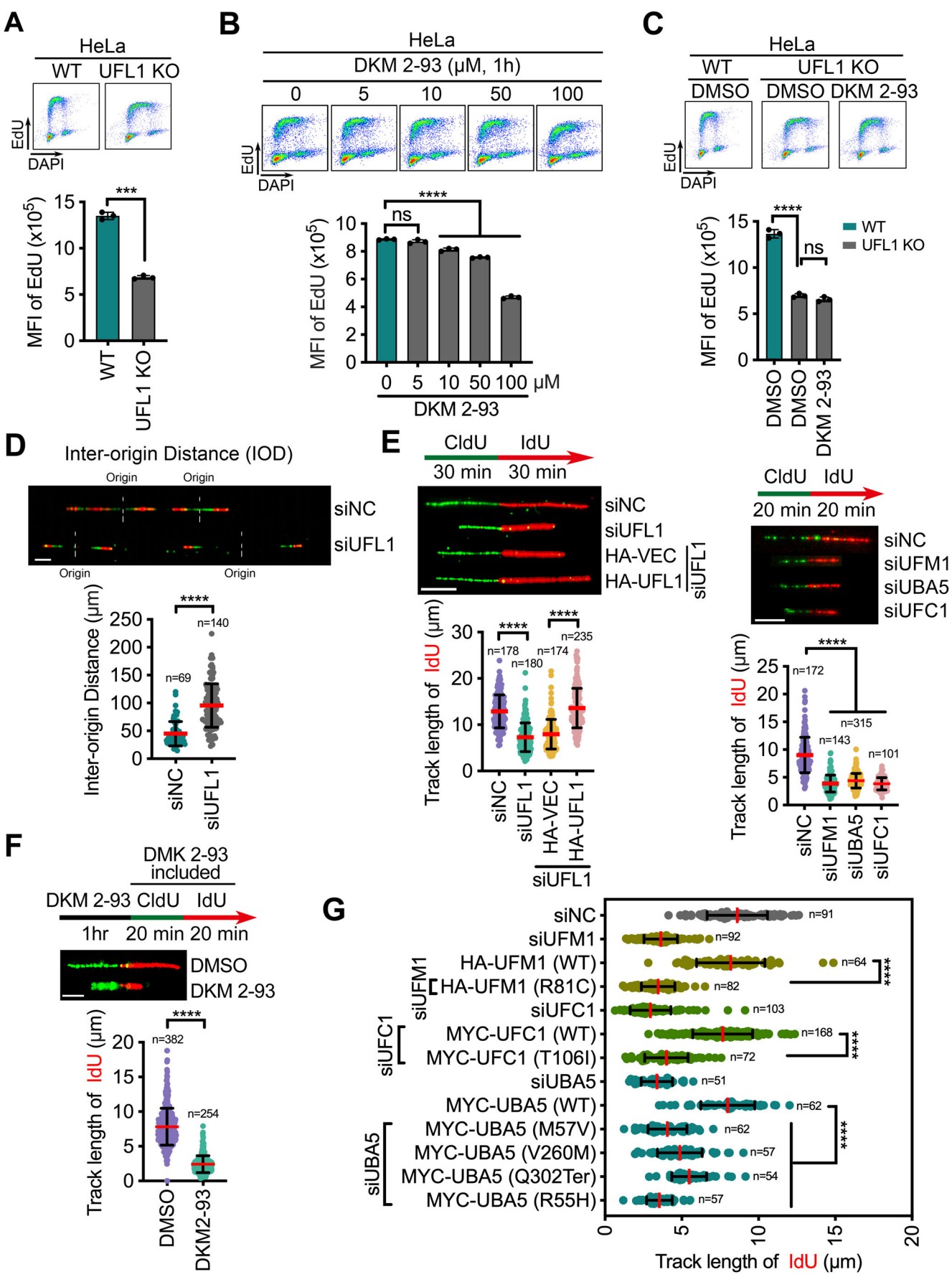

**Figure 1. UFMylation is required for DNA replication.**

(A) Quantification of the Mean Fluorescence Intensity (MFI) of EdU in wild-type (*WT*) and UFL1 KO HeLa cells. The *X*-axis represents DAPI staining intensity (DNA content) on a linear scale, and the *Y*-axis represents the EdU uptake intensity on a logarithmic scale. The MFI of EdU in individual cells was monitored by flow cytometry and presented as mean ± SD of three biological replicates ($n = 3$). *P* value was calculated by unpaired *t* test with Welch's correction (***$P < 0.001$). *P* value: 1.46e-004. (B) Quantification of the EdU MFI in HeLa cells pre-treated with the indicated doses of DKM 2-93 for 1 h. Data are presented as mean ± SD of three biological replicates ($n = 3$). *P* values were calculated by Ordinary one-way ANOVA (****$P < 0.0001$; ns, no significance). *P* values: 0 μM vs 5 μM, 0.232; 0 μM vs 10 μM, 8.47e-006; 0 μM vs 50 μM, 3.75e-008; 0 μM vs 100 μM, 3.85e-013. (C) Quantification of the EdU MFI in HeLa *WT* and UFL1 KO cells pre-treated with DKM 2-93 (100 μM, 1 h). Data are presented as mean ± SD of three biological replicates ($n = 3$). *P* values were calculated by Ordinary one-way ANOVA (****$P < 0.0001$; ns $P > 0.05$, no significance). *P* values: WT DMSO vs UFL1 KO DMSO, 4.86e-007; UFL1 KO DMSO vs UFL1 KO DKM 2-93, 0.326. (D) Quantifications of the inter-origin distances (IOD) in siNC or siUFL1 cells. The top panel shows a representative fiber image, and the lower panel shows the quantification data. Cells were sequentially labeled with CldU and IdU for 20 min each. Green tracts, CldU; red tracts, IdU. Data are presented as mean ± SD. *n* DNA fiber number. *P* value was calculated by unpaired *t* test with Welch's correction (****$P < 0.0001$). *P* value: 1.87e-025. Scale bar, 10 μm. (E) Quantification of the IdU track length in HeLa cells transfected with the indicated siRNAs and plasmids (empty vector [HA-VEC] and HA-UFL1 expression vector [HA-UFL1]). Cells were sequentially labeled with CldU and IdU for indicated time. Data are presented as mean ± SD. *n* DNA fiber number. *P* values were calculated by Ordinary one-way ANOVA (****$P < 0.0001$). *P* values of left panel: siNC vs siUFL1, 1.85e-042; siUFL1+HA-VEC vs siUFL1+HA-UFL1, 1.00e-047. *P* values of right panel: siNC vs siUFM1/siUBA5/siUFC1, 6.97e-091/2.56e-101/2.66e-078. Scale bar, 5 μm. (F) Quantification of the IdU track length in HeLa cells pre-treated with DKM 2-93 (100 μM) for 1 h before CldU and IdU labeling in the presence of the DKM 2-93 for 30 min. Data are presented as mean ± SD. *n* DNA fiber number. *P* value was calculated by unpaired *t* test with Welch's correction (****$P < 0.0001$). *P* value: 8.27e-142. Scale bar, 3 μm. (G) The rate of replication in cells expressing the mutant UFM1, UBA5, and UFC1 genes that cause MPD disorders. Data are presented as mean ± SD. *n* DNA fiber number. *P* value was calculated by Ordinary one-way ANOVA (****$P < 0.0001$). *P* values: HA-UFM1 (WT) vs HA-UFM1 (R81C), 2.6e-064; MYC-UFC1 (WT) vs MYC-UFC1 (T106I), 1.01e-055; MYC-UBA5 (WT) vs MYC-UBA5 (M57V/A260T/Q302Ter*/R55H), 6.49e-041/5.37e-026/7.63e-017/4.24e-049. Source data are available online for this figure.

defects in DNA replication, with mutations in 13 different replication genes, including the components of replicative DNA helicase: *MCM3*, *MCM5*, *MCM7*, *CDC45*, *GINS2*, *GINS3*, and *DONSON* (Nielsen-Dandoroff et al, 2023; Stewart 2024). Regarding the latter, recent research identified that mutations in DONSON associated with MPD might affect its dimerization (Cvetkovic et al, 2023). Interestingly, germline mutations in genes encoding UFM1, UBA5, and UFC1 have also been linked with MPD disorders (Colin et al, 2016; Duan et al, 2016; Muona et al, 2016; Hamilton et al, 2017; Nahorski et al, 2018). These findings led us to hypothesize that UFMylation is involved in DNA replication and that defective UFMylation likely causes MPD disorders by dysregulating the CMG helicase.

Here, we aimed to understand how UFMylation is mechanistically linked to DNA replication and why perturbed UFMylation might lead to the development of MPD-like disorders. To do so, we performed a proteome analysis to identify critical substrates of UFMylation likely involved in regulating the replicative DNA helicase complex. In brief, we uncovered that UFMylation of MCM5 at Lys583 regulates the CMG helicase and ensures normal origin firing and replication fork progression. Moreover, mutations in UFMylation enzymes that cause hereditary microcephaly delay the rate of DNA replication. Together, our data suggest that MCM5 UFMylation ensures the timely completion of DNA replication and genome integrity.

## Results

### UFMylation promotes DNA replication

In our first set of analyses, we wanted to understand how UFMylation controls the cell cycle. To do so, we first confirmed the impact of UFMylation deficiency on cellular proliferation. We prevented UFMylation by three approaches: siRNA-mediated depletion (siUFL1), knockout of the *UFL1* gene (UFL1 KO), and chemical inhibition of the UBA5 E1 UFMylation enzyme by DKM 2-93 treatment (Roberts et al, 2017). Consistent with previous studies [8, 12, 14, 15], all three approaches decreased the proliferation rate of HeLa cells (Fig. EV1A–E). Next, we investigated the mechanistic underpinnings of this observed proliferation defect. By cell cycle analysis, we saw that

UFL1 KO HeLa cells exhibited a 50% reduction in the uptake of pulse-labeled EdU in S-phase cells (Fig. 1A), implying a role for UFMylation in DNA replication. DKM 2-93 exposure also suppressed EdU incorporation in UFMylation-proficient HeLa (Fig. 1B), U2OS (Fig. EV1F), and A549 (Fig. EV1G) cancer cells, 1 h after exposure. As the DKM 2-93 treatment had no detectable effect on DNA replication in UFL1 KO cells (Fig. 1C), we could exclude the off-target effects of this inhibitor. The early suppression of DNA replication by DKM 2-93 in these three cancer cell lines led us to hypothesize that UFMylation of replisome components might control DNA replication.

### UFMylation controls the number and the progression rate of replication forks

The extent of EdU uptake can be used to infer the number of replication forks and the progression rate of individual replication forks. To generate this information, we measured the distance between two nearby fired origins (known as the inter-origin distance, IOD) and the percentage of red-green-red tracks (RGR, new fired origins during CldU labeling) relative to the total number of red-green tracks (RG) in HeLa cells by using DNA fiber assay. UFL1 depletion by siRNA (siUFL1) significantly increased the IOD and decreased the RGR/RG ratio (Figs. 1D and EV1H), indicating a reduction in the number of replication forks in these cells. These data led us to posit that UFMylation might promote origin firing.

Next, we performed a DNA fiber assay to analyze fork progression dynamics. To this end, we sequentially labeled newly synthesized DNA in HeLa cells with two nucleoside analogs, CldU and then with IdU. By measuring the IdU track length, we could estimate replication speed. The IdU track length in siUFL1 cells was ~50% shorter than in negative control (siNC) cells, while UFL1 re-expression fully reversed this defect in three UFL1-depleted cancer cell lines (Fig. 1E, Left; Fig. EV1I). Separate depletion of UFM1, UBA5, and UFC1 by siRNA also reduced the rate of DNA replication by 2.5-fold (Fig. 1E, Right and Fig. EV1J), as did 1 h exposure to DKM 2-93 by 3.5-fold in all three cancer cell lines (Figs. 1F and EV1K,L). These data suggested that constitutive UFMylation activity is required to maintain a normal fork progression rate.

Table 1. Mutations of UFMylation factors in MCPH.

| Genes | Gene mutation | Amino acid change | References |
|---|---|---|---|
| UFM1 | c.241C>T | R81C | (Nahorski et al, 2018) |
| UBA5 | c.169A>G | M57V | (Colin et al, 2016; Muona et al, 2016) |
| | c.164G>A | R55H | |
| | c.778G>A | V260M | |
| | c.904C>T | A302Ter | |
| UFC1 | c.317C>T | T106I | (Nahorski et al, 2018) |

The observed reductions in replication speed might have occurred due to the accumulation of stalled replication forks and/or activation of the replication checkpoint (Saxena and Zou, 2022). We thus monitored the activity of the replication checkpoint and saw no increase in γH2AX or CHK1 phosphorylation at Ser-345 in cells with DKM 2-93 for 1 h (Fig. EV1M), confirming that the slowed replication does not result from replication stress or DNA damage. These findings suggest that UFMylation controls the rate of physiological DNA replication, possibly by UFMylating components of the replisome.

## MPD-causative mutations in *UFM1*, *UBA5*, and *UFC1* impair DNA replication

Germline mutations in genes encoding CMG helicase components (including *MCM3*, *MCM5*, *MCM7*, *CDC45*, *GINS2*, and *GINS3* (Nielsen-Dandoroff et al, 2023; Stewart 2024)) and in genes encoding UFMylation components (*UFM1, UBA5,* and *UFC1*) cause MPD disorders (Colin et al, 2016; Duan et al, 2016; Muona et al, 2016; Hamilton et al, 2017; Nahorski et al, 2018). We hypothesized, therefore, that the known mutations in *UFM1*, *UBA5*, and *UFC1* likely dysregulate DNA replication. To test this hypothesis, we depleted UFM1, UBA5, and UFC1 proteins by siRNA (Fig. 1E, Right) and reconstituted the resulting cells by over-expressing *UFM1*, *UBA5*, and *UFC1 WT* or mutant genes that cause MPD disorders (Table 1), respectively. Results of a DNA fiber analysis showed that only the ectopic expression of the *WT UFM1* transgene in UFM1-depleted cells normalized the rate of replication; none of the mutant *UFM1* transgenes had this effect (Figs. 1G and EV1N). We obtained similar results in the context of *UBA5* and *UFC1* (Figs. 1G and EV1O,P). These data support the idea that germline MPD mutations in *UFM1*, *UBA5*, and *UFC1* result from the dysregulation of the CMG helicase.

## UFL1 is present at replication origins and ongoing replisome

Having confirmed that DNA replication is controlled, in part, by UFMylation events, we next explored the specific role of UFMylation in this process. First, we analyzed the association of UFMylation E3 ligase complex UFL1-UFBP1-CDK5RAP3 (Makhlouf et al, 2024; DaRosa et al, 2024), and the main de-UFMylase UFSP2 (Ishimura et al, 2017) with genomic DNA during the cell cycle. To this end, we synchronized *WT* HeLa cells by two methods: nocodazole block to enrich cells at prometaphase before subsequent release (Fig. EV2A,B) and double-thymidine (dT) block to enrich cells at the G1/S boundary before release (Fig. EV2C,D). By both methods, we confirmed the presence of UFL1 in the chromatin fraction during the cell cycle.

We then analyzed the association of UFL1 with the replisome by "isolating proteins on nascent DNA" (iPOND) assay. The data indicated that endogenous UFL1 was present on EdU-pulse-labeled nascent DNA but not on EdU-labeled DNA after pulse-chase with thymidine (Fig. 2A). We confirmed these findings by proximity ligation assay (PLA). Again, the pulse-label with EdU induced >10 UFL1/EdU PLA signals per cell, and this value dropped to three per cell in the context of the pulse-chase with thymidine (Fig. EV2E,F). These findings suggest that UFL1 is present at the replication fork, agreeing with our earlier data that chemical inhibition of UBA5 restrains replication speed at an early point (Figs. 1F and EV1K,L).

We next asked whether UFL1 is associated with replication origins. We performed chromatin immunoprecipitation (ChIP) assays in HeLa cells at the well-studied *LMNB2* and *MYC* origins and verified UFL1 association (Fig. 2B) along with the known origin-binding protein ORC1 (Costa and Diffley, 2022). We thus conclude that UFL1 is present at two identified replication origins and ongoing replisomes.

## UFMylation affects the timing of the origin firing and progression of the S phase

The association of UFL1 at replication origins led us to investigate the role of UFMylation in origin firing. Loss of UFL1 in UFL1 KO HeLa cells increased the percentage of G1-phase cells by 25% (Fig. EV2G), suggesting a delay in S-phase entry. To verify this idea, we synchronized cells by releasing them from a nocodazole block (time zero). Subsequent cell cycle progression analysis showed that siNC and siUFL1 cells entered the S phase at 6 and 8 h, respectively (Fig. 2C,D). Exposing *WT* HeLa cells to DKM 2-93 also delayed S phase entry by the same degree (Fig. 2E,F). We conclude, therefore, that UFMylation facilitates replication origin firing to initiate the S phase.

We also synchronized cells using the dT block method (Fig. 2G). In this case, DKM 2-93 did not delay the initiation of DNA replication after release from the dT block (Fig. 2H), presumably because the prolonged block at the G1/S boundary for 18 h (Fig. 2G) allows for the assembly of a functional CMG helicase complex at early replication origins. As expected from our earlier identification of decreased EdU uptake following DKM 2-93 exposure (Fig. 1B), however, these cells showed a delay in the completion of the S phase by ~2 h (Fig. 2H). These data suggest that UFMylation is required not only for early replication origin firing but also for efficient replication progression during the S phase.

## UFL1 binds to CMG helicase components

Knowing that UFL1 is present in the chromatin fraction, we next analyzed proteins that physically interact with UFL1. We performed whole-cell immunoprecipitation (IP) in FLAG-UFL1 expressed HEK293T cells using anti-FLAG antibody-conjugated M2 beads (hereafter called M2 Beads) and identified associated partner proteins by mass spectrometry (Fig. 3A). As expected from the FLAG-UFL1 immunocomplex, we identified histone H4 (Fig. 3B), a known interactor of UFL1 (Qin et al, 2019), as well as the UFMylation enzymes UFC1 and UFM1 (Fig. 3B). Interestingly, we also identified several MCMs, namely MCM3, MCM5, and MCM7 (Fig. 3B), suggesting an interaction between UFL1 and the CMG helicase complex. To validate this concept, we conducted

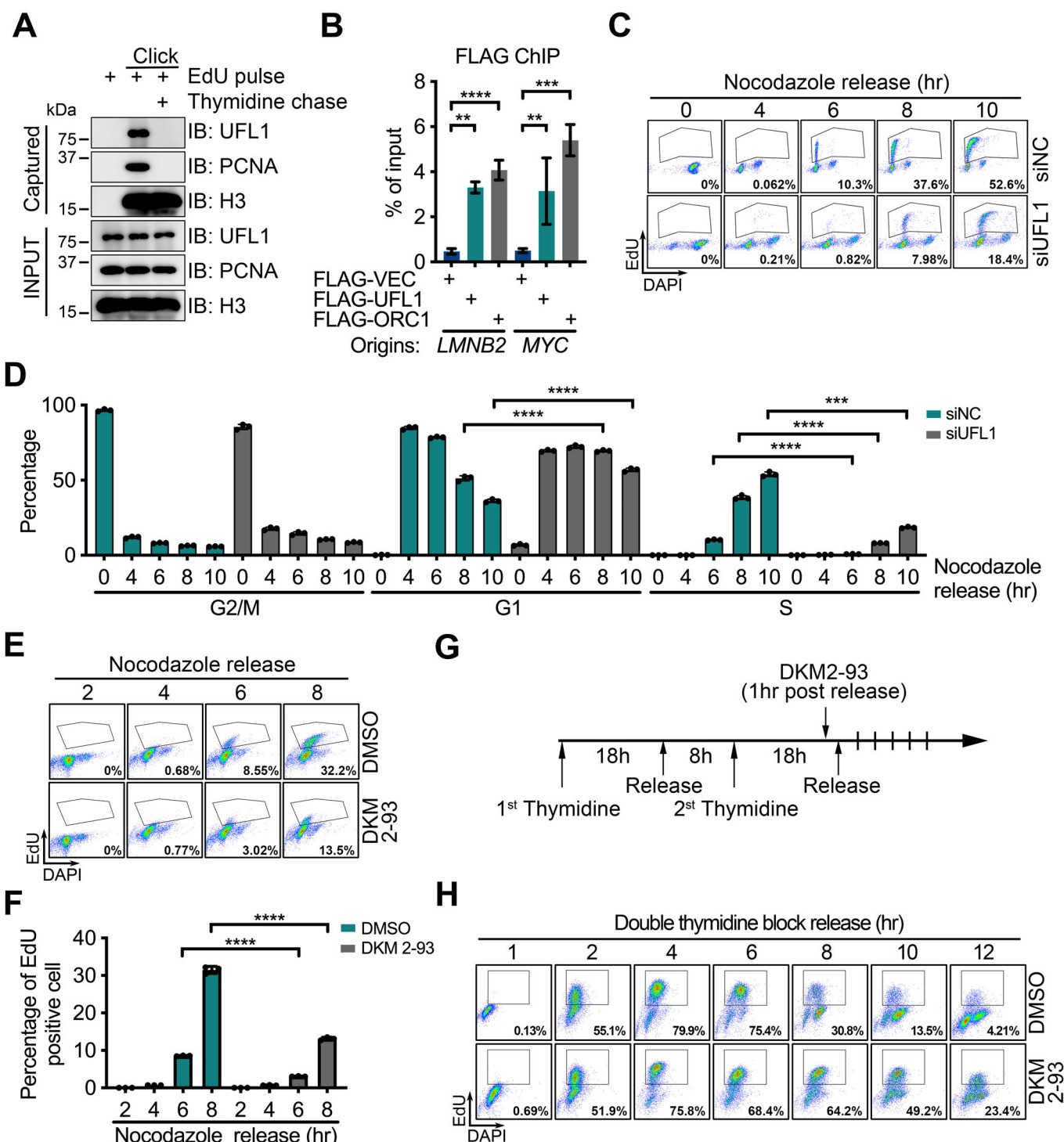

endogenous co-IP assays and indeed detected the CMG helicase subunits MCM2, MCM4, MCM5, GINS3, and CDC45 in the UFL1 immunocomplexes (Fig. 3C). To confirm the interaction between UFL1 and these CMG helicase components in situ, we performed a PLA. The results revealed that UFL1 interacts with helicase complex components MCM5, GINS3, and CDC45 (Fig. EV3A,B), supporting the notion that the CMG helicase can interact with UFMylation machinery.

## UFL1 facilitates replication origin firing by promoting GINS complex and CDC45 loading onto the MCM hexamer

To explore the mechanism by which UFMylation promotes DNA replication, we measured the chromatin loading of proteins implicated in origin firing. Depleting UFL1 had no notable effect on MCM5 loading but resulted in a ~50% decrease in the

chromatin loading of CDC45, GINS3, and PCNA (Fig. EV3C). Similarly, exposure to DKM 2-93 also reduced the chromatin loading of GINS2, GINS3, and CDC45 in a dose-dependent manner, but again, not MCM5 or MCM2 (Fig. EV3D). These data suggest that UFMylation plays a role in the replisome. Mechanistically, UFMylation might facilitate the loading of the GINS complex and CDC45 onto the MCM hexamer or might stabilize the CMG helicase complex. These possibilities are not mutually exclusive.

**Figure 2. UFL1 is present at replication origins and affects their timing.**

We then performed ChIP assays to measure the loading of these factors at the replication origin. Depletion of UFL1 resulted in a decrease in the association of GINS3 and CDC45 at the *LMNB2* origin (Fig. 3D) without affecting the association of ORC1 and MCM7 (Fig. 3E). These data thus pinpointed that UFMylation facilitates CDC45 and GINS complex loading onto replication origins.

To clarify the timing of UFL1 recruitment to replication origins, we performed a ChIP-qPCR assay to examine its dynamic binding across different cell cycle phases, including G1, G1/S, S, G2, and M (Fig. 3F). The data indicated that UFL1 is recruited onto origin at G1/S boundary.

Next, we investigated the temporal dynamics of replication factor binding to chromatin during the cell cycle. In nocodazole-synchronized HeLa cells, we saw that the expected loading of MCM proteins onto chromatin preceded that of GINS3 and GINS4 at 6 h (Figs. 3H and EV3E), coinciding with the initiation of DNA replication (Fig. 2D,F). Notably, treating these cells with DKM 2-93 or siUFL1 delayed the loading of GINS3 by 2 h compared to cells treated with solvent or siNC (Figs. 3F and EV3E). This delay also coincided with the retarded initiation of DNA replication during treatment with siUFL1 or DKM 2-93 (Fig. 2D,F). We thus posit that UFMylation facilitates the loading of CDC45 and GINS onto MCM hexamers to form a functional CMG helicase that can stimulate origin firing.

## UFL1 associates with MCM5 and constitutively UFMylates it

As UFL1 physically interacts with MCMs (Fig. 3B,C), we tested whether MCM2-7 is UFMylated. We transiently expressed FLAG-tagged MCM2-7 individually in HEK293T cells, along with the UFMylation enzymes. We then analyzed the UFMylation of individual MCM components and observed a strong HA-UFM1 signal only for MCM5 with a slow migration pattern on the gel

(Fig. 4A). We also saw stable complex formation between purified MCM5 and UFL1 (Fig. 4B). These data suggest that UFL1 interacts with MCM5 directly to UFMylate it. To test this concept, we lysed cells expressing HA-UFM1 under denaturing conditions, purified endogenous MCM5 by IP, and assessed the covalent binding of UFM1 to MCM5. Western blotting with anti-HA-UFM1 and anti-MCM5 antibodies detected slowly migrating bands corresponding to UFMylated MCM5 (Fig. 4C). Treatment of these cells with siUFL1 or DKM 2-93 reduced the intensity of the slowly migrated bands (Fig. 4D,E), confirming that UFL1 UFMylates MCM5.

We next tested whether MCM5 UFMylation is reversible. To do so, we expressed a MYC-tagged UFM1-specific peptidase 2 (UFSP2) (Komatsu et al, 2024) construct together with FLAG-tagged MCM5 and HA-tagged UFM1 (delC2). Over-expressing MYC-UFSP2, but not a catalytic-dead MYC-UFSP2 (C302S) variant, moderately decreased MCM5 UFMylation levels (Fig. 4F). We thus conclude that MCM5 is a substrate for UFL1 and UFSP2 and that the activity of these two enzymes determines the level of UFMylated MCM5.

## The UFMylation of MCM5 occurs at the ongoing replication fork

Next, we wanted to understand whether MCM5 is UFMylated by UFL1 in the replisome. To address this question, we expressed FLAG-MCM5 and HA-UFM1 in HeLa cells, then isolated and denatured the cytoplasm, nucleoplasm, and chromatin fractions and subjected them to IP using FLAG M2 beads. Here, we learned that MCM5 is UFMylated exclusively in the chromatin fraction (Fig. EV4A).

To determine the timing of MCM5 UFMylation during the cell cycle, we again synchronized cells using nocodazole. When the cells started DNA replication 8 h after release from the nocodazole block (see Fig. EV2B), UFL1 was associated with MCM5 (Fig. EV4B) and UFMylated MCM5 became detectable (Fig. EV4C). These data imply that MCM5 UFMylation occurs at functional replication origins and forks.

To confirm this hypothesis, we purified the active replisome and quantified the extent of UFMylated MCM5. As there is currently no gold-standard method, we employed two techniques to purify the active replisome and checked the reproducibility of the results (Fig. EV4D,F). By the first method, we purified the replisome by

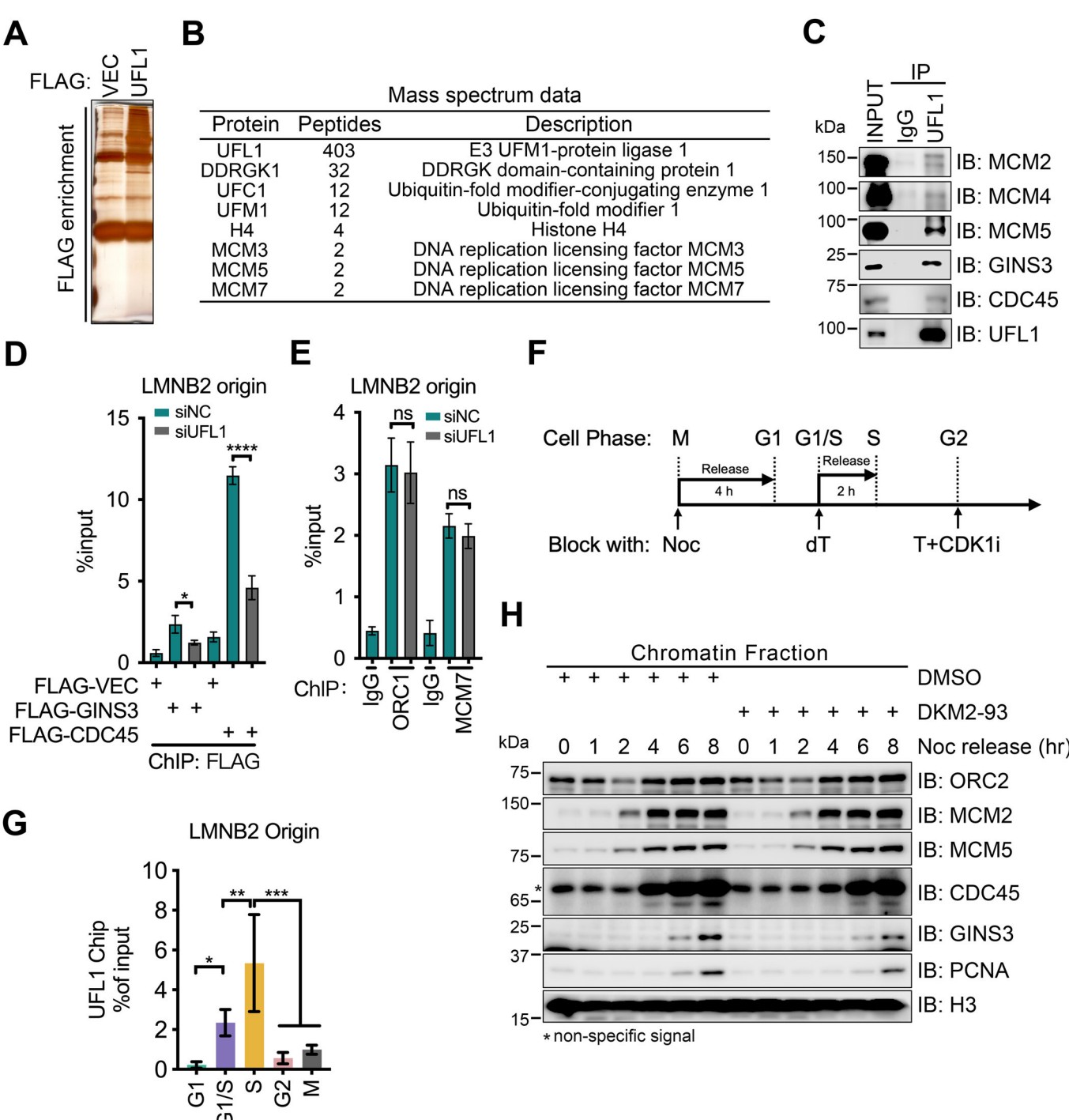

labeling nascent DNA with EdU and purifying proteins associated with EdU-labeled DNA before quantifying UFMylated MCM5 (Fig. EV4D). Here, we found UFMylated MCM5 when we pulse-labeled cells with EdU (Fig. EV4E), supporting the idea that MCM5 is UFMylated at the replisome. By the second method, we performed an IP of the chromatin fraction using an anti-CDC45 or GINS3 antibody, from which we subsequently purified MCM5 by performing an IP with an anti-MCM5 antibody and quantified UFMylated MCM5 levels in this second set of immunoprecipitants (Fig. EV4F). A UFM1 signal

was detectable in the CDC45/MCM5 and GINS3/MCM5 immuno-complexes (Fig. EV4G). These two methods comprehensively support the notion that MCM5 is UFMylated on active replisomes.

## MCM5 UFMylation promotes the CMG complex formation

In our next analyses, we investigated the potential regulatory role of UFMylation during CMG helicase formation by monitoring GINS

◄ **Figure 3. UFL1 binds to CMG helicase components and affects their loading to replisome.**

(A) Silver staining of immunoprecipitants on SDS-PAGE. HEK293T cells transfected with control (VEC)- or FLAG-UFL1 were lysed and subjected to immunoprecipitation (IP) with an anti-FLAG antibody. IP products were separated by SDS-PAGE. (B) List of proteins in (A) identified by mass spectrometry. (C) Western blot analysis of proteins in the indicated immunoprecipitants. HEK293T cell lysates were precipitated with an anti-UFL1 antibody or a control (IgG). (D) ChIP analysis to analyze the association of GINS3 and CDC45 at the *LMNB2* replication origin in HeLa cells transfected with siNC- or siUFL1 and the indicated plasmids. Cell lysates were subjected to ChIP assays with a FLAG antibody. The *Y*-axis represents the amount of qPCR products in the IP product relative to that in the input. The data are presented as mean ± SD of three biological replicates ($n = 3$). *P* value was calculated by Ordinary one-way ANOVA (*$P < 0.05$; ****$P < 0.0001$). *P* values of FLAG-GINS3 group: siNC vs siUFL1, 0.0217. *P* values of FLAG-CDC45 group: siNC vs siUFL1, 7.76e-010. (E) ChIP analysis to analyze the association of endogenous ORC1 and MCM7 at the *LMNB2* replication origin in HeLa cells transfected with siNC- or siUFL1. Cell lysates were subjected to ChIP assays with ORC1 or MCM7 antibodies. The data are presented as mean ± SD of three biological replicates ($n = 3$). *P* value was calculated by Ordinary one-way ANOVA (ns $P > 0.05$, no significance). *P* values of ORC1 group: siNC vs siUFL1, 0.864. *P* values of MCM7 group: siNC vs siUFL1, 0.77. (F, G) ChIP analysis to analyze the association of endogenous UFL1 at the *LMNB2* replication origin in HeLa cells across different cell cycle phases. The method of synchronization is shown in (F), and data are presented in (G). For M phase cells, HeLa cells were treated with 333 nM nocodazole for 16 h and collected by shake-off. For G1 phase cells, the mitotic cells were sub-cultured into fresh medium for 4 h. The G1/S transition was achieved via double thymidine block (dT). For S phase cells, HeLa cells blocked by dT were released for 2 h. The G2 phase cells were obtained using a two-step treatment with thymidine (2 mM, 18 h) followed by a CDK1 inhibitor (RO-3306; 10 μM, 16 h). The data are presented as mean ± SD of three biological replicates ($n = 3$). *P* value was calculated by Ordinary one-way ANOVA (*$P < 0.05$; **$P < 0.01$; ***$P < 0.001$). *P* values: G1 vs G1/S, 0.046; G1/S vs S, 9.23e-003; S vs G2, 4.44e-004; S vs M, 8.79e-004. (H) Western blot analysis of the chromatin fractions of HeLa cells that were synchronized with nocodazole for 16 h and released into fresh culture medium containing DKM 2-93 (100 μM) at the indicated time points. Source data are available online for this figure.

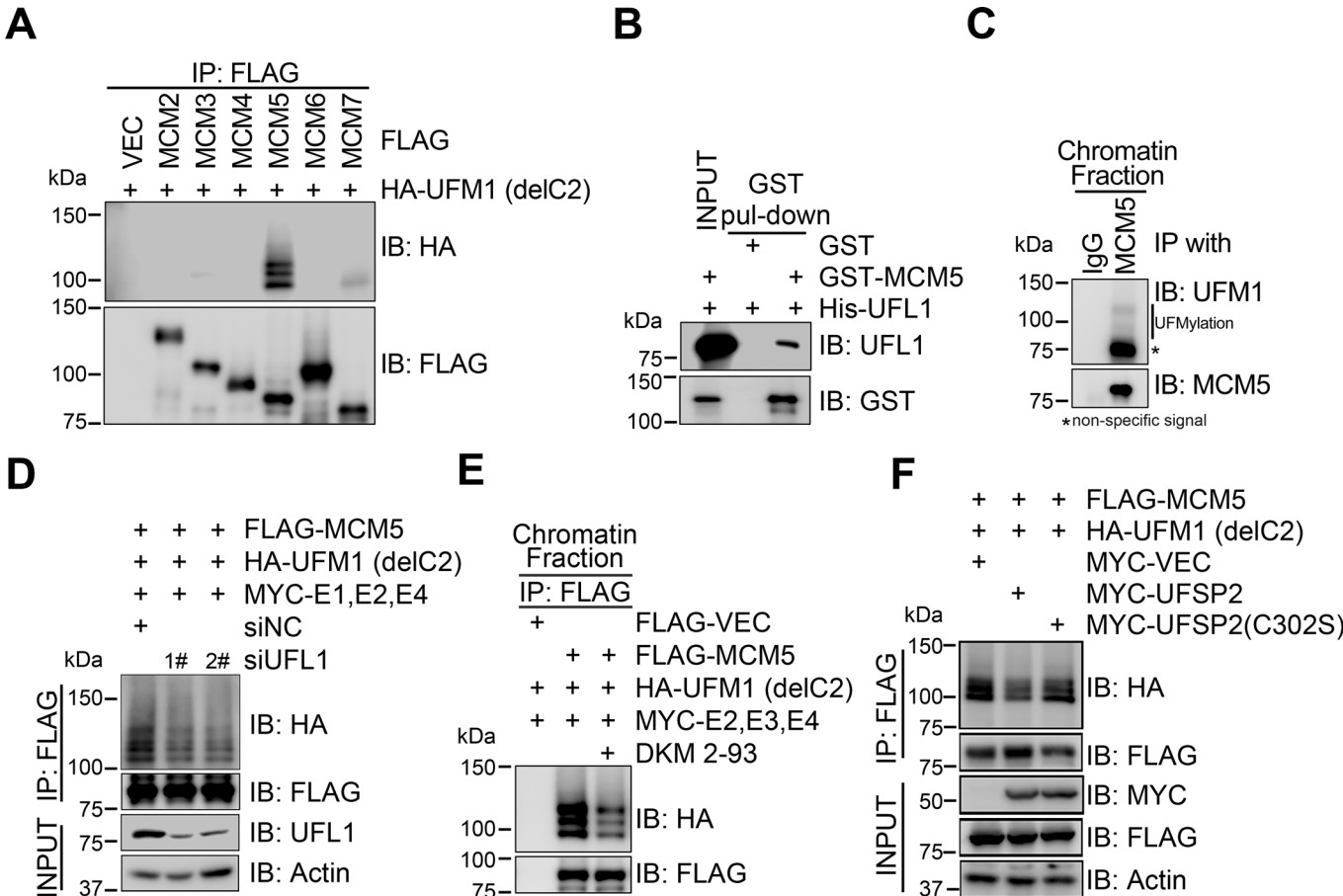

**Figure 4. UFL1 is associated with MCM5 and constitutively UFMylates it.**

(A) UFMylation factors [UBA5, UFC1, UFL1, UFBP1, HA-UFM1 (delC2)] were transiently expressed in HEK293T cells together with either FLAG-MCM2-7 or FLAG-tag alone (VEC), before IP with FLAG M2 beads and western blotting with the indicated antibodies. (B) GST pull-down assay in vitro to detect direct interactions between purified GST-MCM5 and His-UFL1. Proteins associated with the beads were analyzed by western blotting. (C) The chromatin fraction of HeLa cells was denatured and subjected to IP with an anti-MCM5 antibody or IgG before western blot analysis. (D) HEK293T cells expressing the indicated proteins were transiently transfected with siNC- or siUFL1, lysed under denaturing conditions, and subjected to IP with FLAG M2 beads. Immunoprecipitants were analyzed by western blotting with the indicated antibodies. (E) HeLa cells expressing the indicated plasmids were treated with DKM 2-93 (100 μM) for 1 h before cell lysis. The chromatin fraction was isolated and subjected to IP with FLAG M2 beads, followed by western blot analysis. (F) Cells expressing UFMylation factors and MCM5 were transiently transfected with either MYC-VEC, MYC-UFSP2, or MYC-UFSP2 (C302S). MCM5 was purified by IP with anti-FLAG-MCM5 and analyzed by western blotting. Source data are available online for this figure.

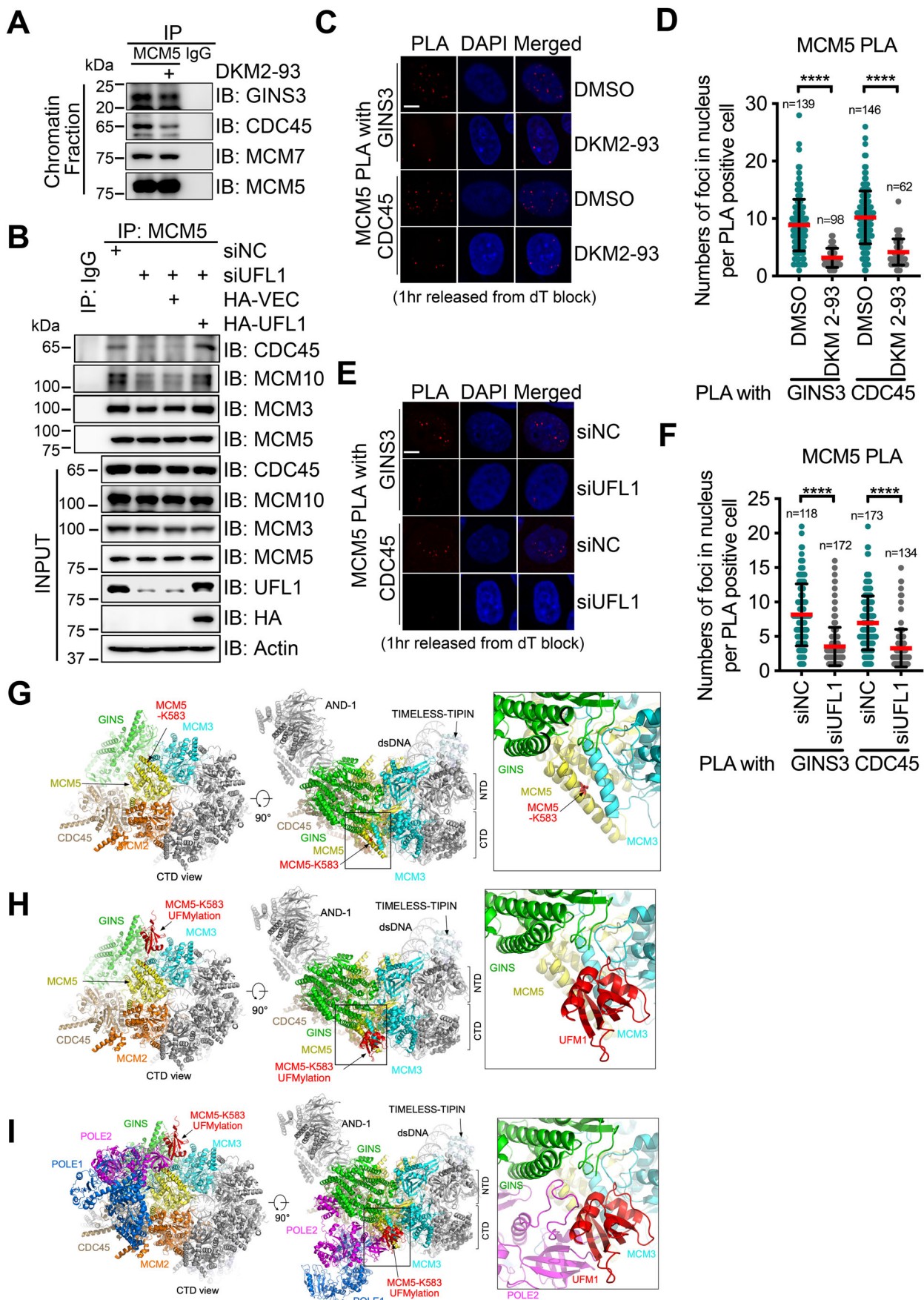

(1hr released from dT block)

Figure 5.   MCM5 UFMylation promotes origin firing.

(A) Western blot analysis of proteins in immunoprecipitants derived from chromatin fractions of HeLa cells treated with DMSO or DKM 2-93 (100 µM). Anti-IgG or anti-MCM5 antibody was used for immunoprecipitation. (B) Western blot analysis of proteins in immunoprecipitants of HeLa cells transfected with siNC or siUFL1. Immunoprecipitants were treated as in (A). (C, D) Immunostaining of PLA foci in HeLa cells at 1 h after release from dT block in the presence of DMSO or DKM 2-93 (100 µM) from 1 h before the release from dT block. Representative images of PLA foci (red) are shown in (C), and the quantitative data are presented in (D). n cell number. P values was calculated by Ordinary one-way ANOVA (****$P < 0.0001$). P value of GINS3 group: DMSO vs DKM 2-93, 7.57e-026. P value of CDC45 group: DMSO vs DKM 2-93, 2.42e-022. Scale bar, 5 µm. (E, F) Immunostaining of PLA foci in HeLa cells transfected with siNC or siUFL1 at 1 h released from dT block. Representative images of PLA foci (red) are shown in (E), and the quantitative data are presented in (F). n cell number. P values was calculated by Ordinary one-way ANOVA (****$P < 0.0001$). P value of GINS3 group: siNC vs siUFL1, 2.65e-025. P value of CDC45 group: siNC vs siUFL1, 4.87e-018. Scale bar, 5 µm. (G) Top and side views of the replisome structure (PDB: 7PFO) displayed in the cartoon representation. The magnified view of the boxed region emphasizes the CTD interface between MCM5 and MCM3. Polε is not shown to clearly illustrate the UFMylation site K583 on MCM5, highlighted in red. (H, I) The view is as shown in (G) but includes UFM1 modeled at MCM5-K583. Polε is displayed in (I) to indicate the location of MCM5 UFMylation relative to the docking site of POLE2 on the MCM2-7 ring within the replisome. Source data are available online for this figure.

and CDC45 loading onto the MCM hexamer. To do so, we treated cells with DKM 2-93 or siUFL1 at 1 h after release from the dT block (see Fig. EV2D). We then examined GINS and CDC45 loading onto the hexamer by analyzing the MCM5 immunocomplexes (Fig. 5A,B) and measured the number of MCM5/GINS3 and MCM5/CDC45 PLA foci per cell (Fig. 5C–F). We saw significant reduction in GINS3 and CDC45 protein levels in MCM5 immunocomplexes isolated from DKM 2-93-treated (Fig. 5A) or siUFL1-treated HeLa cells (Fig. 5B) compared to untreated control cells. The interaction between MCM5 and MCM10, another important factor in origin firing, was also reduced in siUFL1-depleted HeLa cells (Fig. 5B). We also saw decrease in the levels of these PLA foci in DKM 2-93-treated (Fig. 5C,D) and siUFL1-treated (Fig. 5E,F) cells. These findings support that MCM5 UFMylation promotes GINS and CDC45 loading onto the MCM hexamer to form a functional CMG helicase.

## MCM5 is UFMylated at K583

To identify which region of MCM5 is UFMylated, we generated a series of MCM5 deletion mutant cDNAs (Fig. EV5A) and transfected them into cells expressing all UFMylation cascade proteins. We found that MCM5 UFMylation was reduced two-fold in the D6M mutant, which lacked amino acids 501–600, compared to WT MCM5 (Fig. EV5B). We then mutated individual lysine residues within the D6M region and eventually found that a Lys583Arg (K583R) mutation decreased the UFMylation signal both in whole cell lysates and chromatin fraction (Fig. EV5C,D). K583 is a highly conserved site across species (Fig. EV5E), and our analyses suggest it is the primary MCM5 UFMylation site. We confirmed this proposal by using an in vitro UFMylation assay that showed that the UFMylation levels of GST-MCM5 (K583R) were significantly reduced compared to WT GST-MCM5 (Fig. EV5F). When we immunoprecipitated FLAG-tagged MCM5 and immunoblotted IP products by an anti-HA-UFM1, we noted the presence of three bands (Fig. EV5C), implying that both mono- and oligo-UFMylation occurs at MCM5 K583.

In the CMG structure, GINS and CDC45 are firmly anchored onto the N-terminal domain (NTD) ring of MCM2-7 (Jones et al, 2021; Yuan et al, 2016; Rzechorzek et al, 2020). Notably, K583 is situated at the C-terminal domain (CTD) interface between MCM5 and MCM3, a position far removed from the GINS and CDC45 binding sites (Fig. 5G) (Rzechorzek et al, 2020; Jones et al, 2021). We posit, therefore, that it is unlikely that MCM5 UFMylation at

K583 interferes with GINS and CDC45 loading onto MCM2-7 (Fig. 5H,I). This idea agrees with data showing that the MCM5 binding site of UFM1 fits into the gaps between MCM3, GINS, and POLE2, a component of replicative polymerase ε (Fig. 5I). Given that this modification promotes CMG formation, we hypothesize that MCM5 K583 UFMylation might rather serve as a scaffold to facilitate GINS and CDC45 loading onto MCM2-7 or to stabilize transient intermediates by engaging with specific origin firing factors during helicase activation.

## MCM5 UFMylation at K583 promotes origin firing and replication fork progression

Next, we asked whether UFMylation promotes cell proliferation by UFMylating MCM5 at K583. We measured the rate of proliferation and found that MCM5 (K583R)-expressing cells proliferated more slowly than WT MCM5-expressing HeLa cells (Fig. EV6A,B). Because of this result, we investigated the impact of MCM5 K583 UFMylation on DNA replication. The IOD was approximately two-fold larger in MCM5 (K583R)-expressing cells compared to WT MCM5-expressing cells (Fig. 6A,B), similar to our earlier findings on UFL1-depleted cells (Fig. 1D). Thus, an MCM5 (K583R) mutation decreases the number of activated origins. Exposing WT cells to DKM 2-93 also reduced the number of active origins in WT MCM5-expressing cells but not MCM5 (K583R)-expressing cells (Fig. 6A,B). These data indicate that UFL1 promotes replication origin firing by UFMylating MCM5 at K583.

Next, we measured the progression rate of individual DNA replication forks using the DNA fiber assay (Fig. 6C). MCM5 (K583R) expression decreased the replication speed of HeLa cells by two-fold (Fig. 6C,D), consistent with the results obtained from UFL1-depleted cells (Fig. 1E). This reduction in the progression rate likely results from the defective replication of MCM5 (K583R)-expressing cells. Indeed, exposure to DKM 2-93 reduced the rate of replication in WT MCM5-, but not MCM5 (K583R)-expressing cells (Fig. 6C,D). UFL1 thus seems to control origin firing and replication fork progression by UFMylating MCM5 at K583.

To explore the cause of the reduced origin firing, we performed ChIP assays to measure the association of CMG helicase components at the LMNB2 origin in WT MCM5- and MCM5 (K583R)-expressing cells (Fig. 6E,F). MCM5 (K583R)-expressing cells showed >5-fold decrease in GINS3 and CDC45 signals compared with WT MCM5-expressing cells (Fig. 6E). Meanwhile, WT MCM5- and MCM5 (K583R)-expressing cells both showed

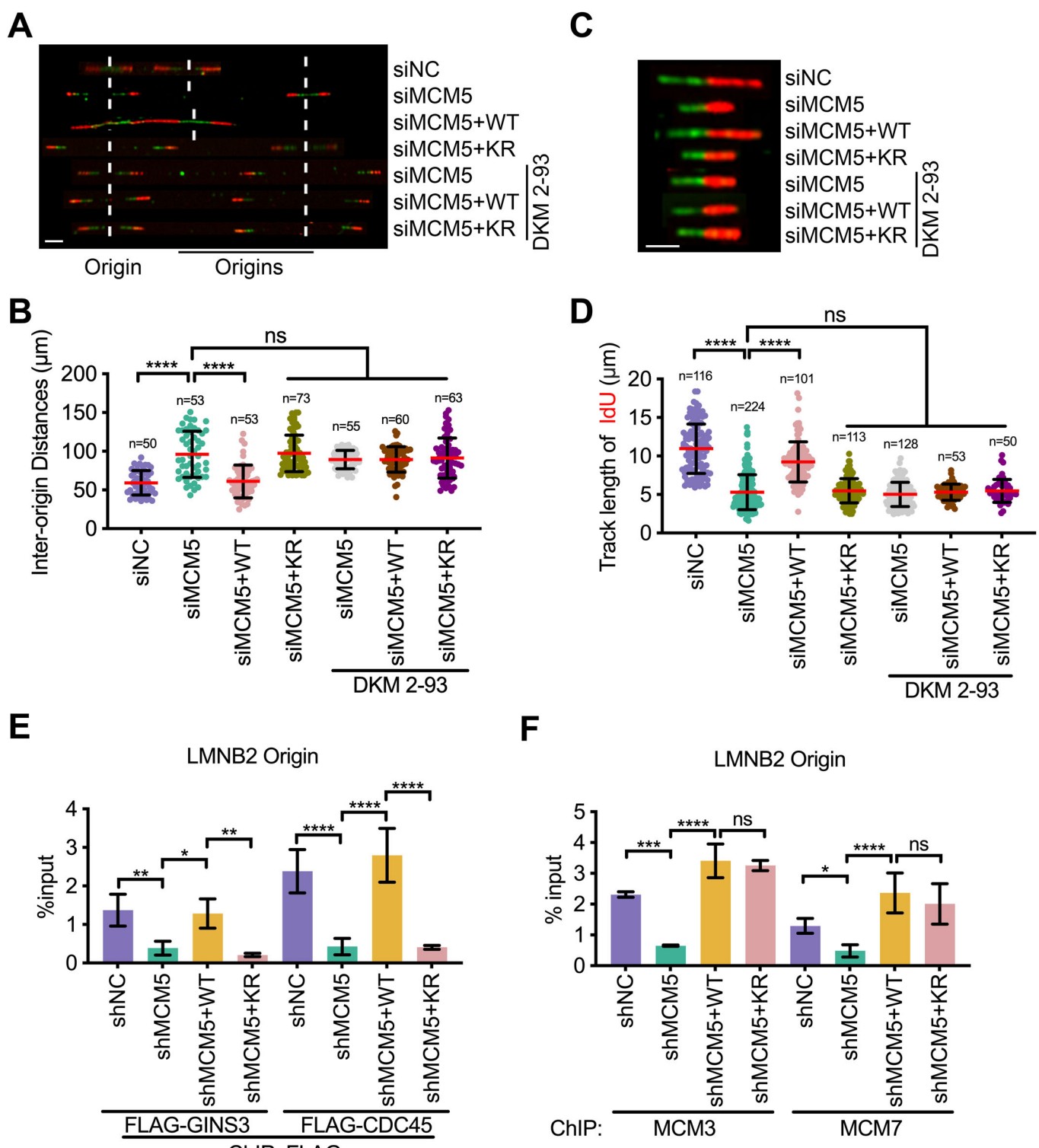

similar levels of ChIP signals relating to MCM components (Fig. 6F). These findings support an essential role for MCM5 K583 UFMylation in promoting GINS and CDC45 loading onto the MCM hexamer at replication origins.

We also investigated the role of MCM5 UFMylation in CMG helicase formation. The quantity of GINS and CDC45 proteins in

the chromatin fraction of MCM5 (K583R)-expressing cells was reduced by ~50% compared to that of *WT* MCM5-expressing cells (Fig. EV6C). We thus performed PLA and IP assays to analyze the ability of GINS and CDC45 to complex with MCM5 (K583R) versus *WT* MCM5. We performed the PLA 1 h after releasing HeLa cells from a dT block to examine CMG helicase complex formation

**Figure 6. MCM5 UFMylation at K583 promotes origin firing.**

(A, B) Quantification of the inter-origin distances in HeLa cells transfected with the indicated siRNAs and plasmids. DKM 2-93 treated cells were pre-treated with DKM 2-93 (100 μM) for 1 h prior to the labeling process, and the drug was maintained throughout both CldU and IdU labeling periods. Representative fiber images are shown in (A), and quantitative data are shown in (B). Green tracts, CldU; red tracts, IdU; dashed line: replication origins. Data are presented as mean ± SD. $n$ DNA fiber number. $P$ value was calculated by Ordinary one-way ANOVA (****$P < 0.0001$; ns $P > 0.05$, no significance). $P$ values: siNC vs siMCM5, 7.64e-016; siMCM5 vs siMCM5+WT, 6.46e-015; siMCM5 vs siMCM5+KR/siMCM5+DKM 2-93/siMCM5+WT + DKM 2-93/siMCM5+KR + DKM 2-93, 0.99/0.457/0.464/0.77. Scale bar, 5 μm. (C, D) Quantification of the IdU track length in the cells used in (A). Individual dots represent the length of each replication fork. Data are presented as mean ± SD. $n$ DNA fiber number. $P$ value was calculated by Ordinary one-way ANOVA (****$P < 0.0001$; ns $P > 0.05$, no significance). $P$ values: siNC vs siMCM5, 1.51e-085; siMCM5 vs siMCM5+WT, 1.58e-043; siMCM5 vs siMCM5+KR/siMCM5+DKM 2-93/siMCM5+WT + DKM 2-93/siMCM5+KR + DKM 2-93, 0.983/0.811/0.999/0.997. Scale bar, 5 μm. (E) ChIP analysis to analyze the association of FLAG-CDC45 and FLAG-GINS3 at the LMNB2 origin in siRNA-treated HeLa cells transfected with the indicated plasmids. The data are presented as mean ± SD of three biological replicates ($n = 3$). $P$ value was calculated by Ordinary one-way ANOVA (*$P < 0.05$; **$P < 0.01$; ****$P < 0.0001$). $P$ values of FLAG-GINS3 group: shNC vs shMCM5, 6.74e-003; shMCM5 vs shMCM5+WT, 0.0119; shMCM5+WT vs shMCM5+KR, 3.61e-003. $P$ values of FLAG-CDC45 group: shNC vs shMCM5, 1.33e-005; shMCM5 vs shMCM5+WT, 1.31e-006; shMCM5+WT vs shMCM5+KR, 1.17e-006. (F) ChIP analysis to analyze the association of endogenous MCM3 and MCM7 at the LMNB2 origin in shRNA-treated HeLa cells transfected with the indicated plasmids. The data are presented as mean ± SD of three biological replicates ($n = 3$). $P$ value was calculated by Ordinary one-way ANOVA (*$P < 0.05$; ***$P < 0.001$; ****$P < 0.0001$; ns $P > 0.05$, no significance). $P$ values of MCM3 group: shNC vs shMCM5, 1.10e-004; shMCM5 vs shMCM5+WT, 2.75e-007; shMCM5+WT vs shMCM5+KR, 0.646. $P$ values of MCM7 group: shNC vs shMCM5, 0.0245; shMCM5 vs shMCM5+WT, 2.94e-005; shMCM5+WT vs shMCM5+KR, 0.294. Source data are available online for this figure.

at the $G_1$/S boundary. We found a four-fold decrease in the number of CDC45/MCM5 (K583R) PLA foci per cell compared to CDC45/WT MCM5 PLA foci per cell (Fig. EV6D,E). Meanwhile, the K583R mutation decreased the quantity of GINS3 and CDC45 proteins co-immunoprecipitated with MCM5 (Fig. EV6F,G). These data suggest that the K583R mutation destabilizes the physical association between the MCM hexamer, CDC45, and GINS, leading to a decrease in CDC45 and GINS association at replication origins (Fig. 6E) and a subsequent delay in the initiation of DNA replication in synchronized cells (Fig. 2C–F).

## MCM5 UFMylation facilitates cell growth and maintains genome stability

Thus far, we have seen that MCM5 UFMylation promotes DNA replication. Our findings led us to hypothesize that impaired replication might block the completion of DNA replication, leading to the formation of anaphase bridges (Fig. EV7A) and ultimately causing genome instability in cancer cells (Primo and Teixeira, 2020; Al Ahmad Nachar and Rosselli, 2022; Teixeira et al, 2015; Techer et al, 2017). UFL1 depletion or MCM5 (K583R) expression significantly increased the percentage of cells exhibiting chromosome bridges (Figs. 7A,B and EV7B). These data indicate that MCM5 UFMylation facilitates DNA replication and prevents anaphase bridge formation.

DNA breaks in un-replicated regions typically occur at common fragile sites (CFSs) under conditions of replication stress, such as that induced by aphidicolin (APH) treatment (Bhowmick et al, 2023; Branzei and Foiani, 2010). In our final experiments, we asked whether defective MCM5 UFMylation is a source of DNA replication stress that could cause breakage at CFSs. Results of a γH2AX ChIP assay indicated that expressing MCM5 (K583R) in MCM5-depleted cells by siRNA indeed leads to enrichment of γH2AX at a typical CFSs (e.g., the FRA16D locus), even in the absence of APH treatment. Reintroducing WT MCM5 rescued this phenotype, implicating a role for MCM5 UFMylation in preserving genome stability at CFSs (Fig. 7C). Indeed, we saw that the number of chromosome aberrations was two times higher in cells expressing MCM5 (K583R) than in cells expressing WT MCM5 (Figs. 7D and EV7C). Together, these findings support that MCM5 UFMylation has a crucial role in facilitating cell growth and maintaining genome stability.

## Discussion

Germline mutations in genes encoding the three UFMylation enzymes (Colin et al, 2016; Duan et al, 2016; Muona et al, 2016; Hamilton et al, 2017; Nahorski et al, 2018) and components of the replicative DNA helicase (Colin et al, 2016; Duan et al, 2016; Muona et al, 2016; Hamilton et al, 2017; Nahorski et al, 2018) cause MPD disorders. Despite these findings, MPD disorders likely stem from a range of causes, and the specific role of defective UFMylation in causing MPD disorders has remained unclear. Additionally, the involvement of UFMylation in cell proliferation has not been fully defined. Given the link between MPD and mutations in CMG helicase components, we hypothesized that UFMylation controls DNA replication. Through an in-depth series of molecular and genetic experiments, we uncovered that UFMylation is essential for efficient DNA replication and that MCM5, a critical component of the replicative DNA helicase, is a UFMylation target. As such, defective UFMylation delays replication origin firing by 2 h and decreases the progression of the S phase, slowing down cell proliferation (Fig. 7E). All known germline MPD mutations in UFM1, UBA5, and UFC1 genes caused a reduction in the rate of replication fork progression (Fig. 1G), supporting the idea that UFMylation prevents MPD disorders by controlling the CMG helicase (Fig. 7E).

## UFMylation of MCM5 at K583 is required for efficient origin firing and replication fork progression

A specific role for UFMylation in normal, unchallenged DNA replication had not been previously studied, likely because UFMylation is involved in multiple cellular processes (Komatsu et al, 2024), such as ER homeostasis (Li et al, 2018; Ishimura et al, 2023; Mao et al, 2023; Makhlouf et al, 2024; DaRosa et al, 2024). As a result, it has been challenging to separate its functions in DNA replication from those in other processes. We filled this knowledge gap by identifying a crucial UFMylation substrate, MCM5, which allowed us to focus on origin firing and cell cycle progression. We found that MCM5 is specifically UFMylated at K583. Depletion of UFL1 or ectopic expression of a UFMylation-resistant MCM5(K583R) mutant resulted in defective GINS3 and CDC45 recruitment to replication origins, reduced the number of activated

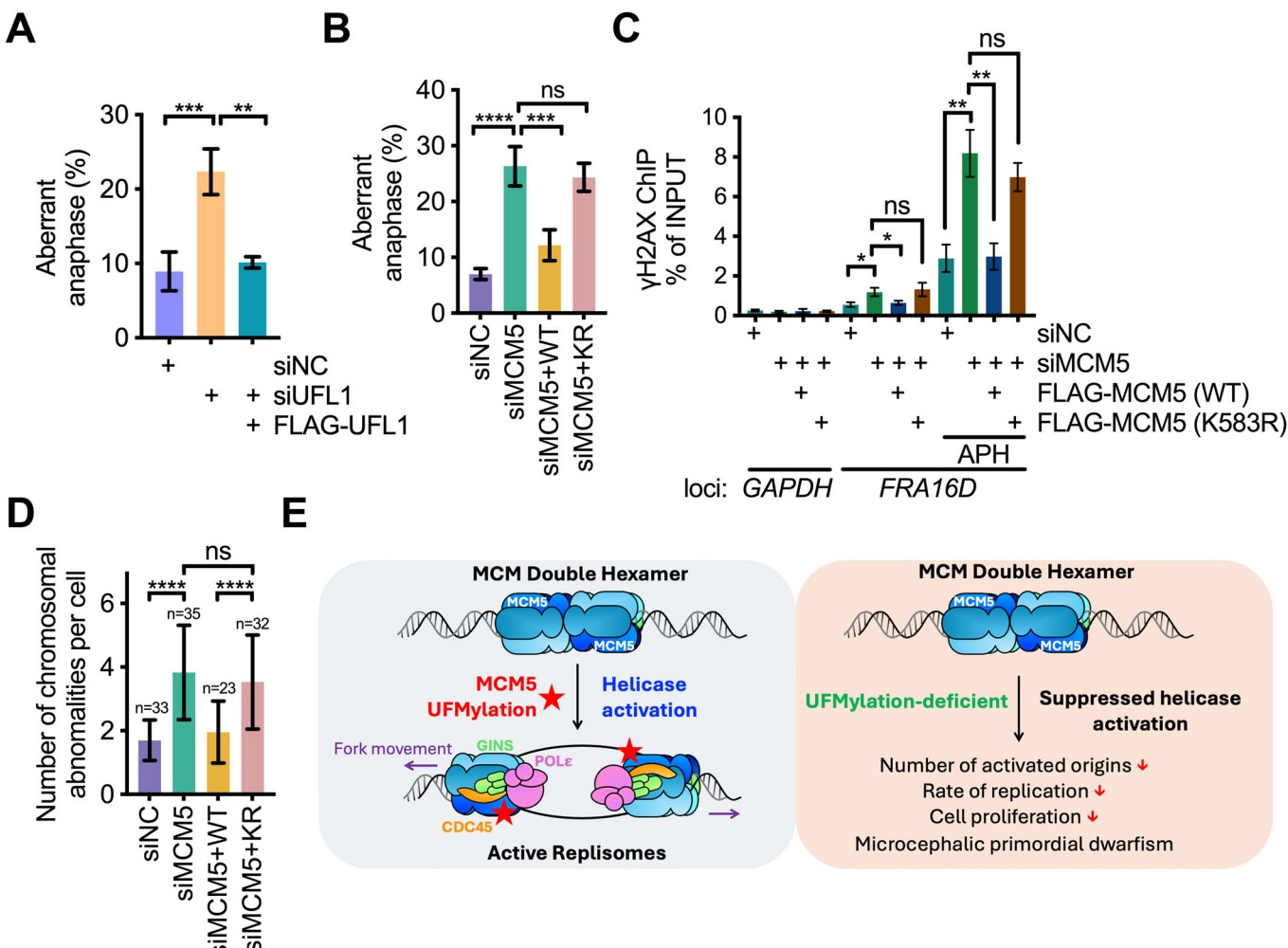

**Figure 7. MCM5 UFMylation promotes cell growth and maintains genome stability.**

(A, B) Quantification of aberrant chromosomes in HeLa cells transfected with the indicated siRNAs and plasmids. The data are presented as mean ± SD of three independent experiments ($n = 3$). *P* value was calculated by Ordinary one-way ANOVA (***P* < 0.01; ****P* < 0.001). *P* values in (A): siNC vs siUFL1, 8.82e-004; siUFL1 vs siUFL1+FLAG-UFL1, 1.48e-003. *P* values in (B): siNC vs siMCM5, 5.24e-005; siMCM5 vs siMCM5+WT, 4.84e-004; siMCM5 vs siMCM5+KR, 0.756. (C) ChIP analysis to analyze γH2AX enrichment at *GAPDH* or *FRA16D* gene loci in HeLa cells transfected with the indicated siRNAs and plasmids. Cells were incubated with or without 0.4 μM APH for 24 h and subsequently cross-linked with formaldehyde and sonicated before IP with a γH2AX antibody. The obtained ChIP DNAs were amplified by qPCR using *GAPDH* or *FRA16D* primers. The data are presented as mean ± SD of three independent experiments ($n = 3$). *P* value was calculated by unpaired *t* test with Welch's correction (**P* < 0.05; ***P* < 0.01; ns *P* > 0.05, no significance). *P* values of *FRA16D* group without APH: siNC vs siMCM5, 0.0179; siMCM5 vs siMCM5+WT, 0.0287; siMCM5 vs siMCM5+KR, 0.6038. *P* values of *FRA16D* group with APH: siNC vs siMCM5, 0.0053; siMCM5 vs siMCM5+WT, 0.0059; siMCM5 vs siMCM5+KR, 0.2204. (D) Metaphase spread of HeLa cells transfected with the indicated siRNAs and plasmids was performed, and the overall number of chromosome abnormalities per cell was determined. The data are presented as mean ± SD. *n* cell number. *P* value was calculated by Ordinary one-way ANOVA (*****P* < 0.0001; ns *P* > 0.05, no significance). *P* values: siNC vs siMCM5, 1.71e-010; siMCM5+WT vs siMCM5+KR, 1.91e-005; siMCM5 vs siMCM5+KR, 0.687. (E) The MCM hexamer (blue) forms a complex with CDC45 (orange) and GINS (green). UFMylation (red pentagram) of MCM5 stabilizes the CMG complex (Left), and a defect in UFMylation reduces the number of active CMG helicases (Right). It remains unclear how UFMylation affects complex formation. Defective UFMylation reduces the efficiency of origin firing and the rate of replication fork progression, leading to impaired proliferation and, eventually, MPD disorders. Source data are available online for this figure.

origins, and caused a 2 h delay in origin firing as well as a two-fold decrease in the progression rate of individual replication forks. The outcome of these events at the cellular level is a two-fold decrease in the DNA replication rate (Fig. 1A–C), indicating that MCM5 UFMylation is essential for effective DNA replication.

Although an MCM5(K583R) mutation could theoretically affect the functionality of the CMG helicase independently of its UFMylation, we consider this possibility unlikely. When we exposed HeLa cells to DKM 2-93 to broadly inhibit UFMylation, we saw a decrease in the number of active origins and the rate of DNA replication in *WT* MCM5-expressing cells. By contrast, these effects with DKM 2-93 were not observed in MCM5 (K583R)-expressing cells, suggesting that the replication defects are likely due to the loss of UFMylation rather than any inherent disruption caused by the mutation.

## The functionality of CMG helicase depends on the UFMylation of MCM5 at K583

Further analyses showed that either a UFL1 deficiency or expression of the MCM5 K583R mutant could weaken the affinity of the MCM hexamer for CDC45 and GINS. Consistently, we saw significant decreases in the number of MCM5/CDC45 and MCM5/GINS foci and the quantity of CDC45 and GINS in the chromatin fraction in the early S phase. The structure of the human replisome also suggests that the MCM5 UFMylation site, K583, is strategically located, potentially playing a critical role in facilitating the assembly of the CMG helicase during helicase activation (Jones et al, 2021). Together, we propose that MCM5 UFMylation at K583 is essential for CMG helicase complex formation at active replication origins and to maintain replisome function.

We postulate that UFMylation-mediated stabilization of the CMG complex increases its melting capability at replication origins and the unwinding kinetics during replication fork progression. This idea is supported by the fact that complex formation involving MCM hexamers with CDC45 and GINS is essential for the functioning of the replicative DNA helicase (Costa et al, 2011; Gambus et al, 2006; Eickhoff et al, 2019; Yuan et al, 2020). If this is the case, we can conclude that MCM5 UFMylation also facilitates origin firing and replication fork progression by controlling the DNA helicase activity of the CMG complex.

## The UFMylation machinery is an integral component of replisome

Through our study, we obtained numerous lines of evidence to support that the UFMylation machinery is present at functional replication origins and replication forks. We showed that: UFL1 localizes with the MCM hexamer, CDC45, GIN3, and nascent DNA (labeled with EdU), preventing UFMylation by UBA5 inhibitor treatment immediately delays replication fork progression; UFL1 and MCM5 directly interact with each other; and that constitutive UFMylation plays a pivotal role in CMG helicase functionality during replication fork progression.

There are other mechanisms that can increase replication speed, such as those based on the extent of ISG15ylation and poly(ADP-ribosyl)ation (PARylation) (Xiang et al, 2023; Maya-Mendoza et al, 2018), but how such mechanisms enhance the activity of the replicative DNA helicase was not previously clarified. We have now demonstrated that MCM5 UFMylation is also required to ensure a normal replication speed. We anticipate that this finding will help with the future elucidation of the molecular mechanisms underlying the response to replication stress caused by base damage on the template DNA strand and oncogenic transformation by analyzing the regulation of UFMylation activity in the replisome.

## UFMylation is a druggable target for treating cancer and MPD

Our findings highlight new directions to explore UFMylation as a target in cancer therapy. Previous studies have identified several small molecules that inhibit MCMs and show promise as anti-cancer agents (Seo and Kang, 2018; Majid et al, 2010; Guan et al, 2017; Mio et al, 2016; Ishimi et al, 2009). Here, we demonstrate that DKM 2-93 significantly reduces the proliferation rate of HeLa, U2OS, and A549 cancer cells by interfering with origin firing and DNA replication progression, suggesting that targeting UFMylation could provide a novel therapeutic strategy. Notably, DKM 2-93 has also been shown to suppress pancreatic cancer cell growth (PaCa2, Panc1) and reduce tumor size in PaCa2 tumor xenograft mouse models (Roberts et al, 2017). Another clinical implication of our study is that deUFMylating enzymes might be druggable targets for preventing MPD associated with replication defects. Together, these findings suggest that enzymes mediating UFMylation could be valuable drug targets for treating certain cancers and MPD.

# Methods

**Reagents and tools table**

| Reagent/resource | Reference or source | Identifier or catalog number |
|---|---|---|
| **Experimental models** | | |
| Trans5a Chemically Competent Cell | TransGen Biotech | Cat #CD201-01 |
| BL21 Chemically Competent Cell | TransGen Biotech | Cat #CD901-02 |
| HeLa (*Homo sapiens*) | ATCC | Cat # CCL-2 |
| U2OS (*Homo sapiens*) | ATCC | Cat # HTB-96.NM |
| A549 (*Homo sapiens*) | ATCC | Cat # CCL-185 |
| HEK293T (*Homo sapiens*) | ATCC | Cat # CRL-3216 |
| **Recombinant DNA** | | |
| pcDNA 3.0 FLAG-MCM2 | This paper | N/A |
| pcDNA 3.0 FLAG-MCM3 | This paper | N/A |
| pcDNA 3.0 FLAG-MCM4 | This paper | N/A |
| pcDNA 3.0 FLAG-MCM5 | This paper | N/A |
| pcDNA 3.0 FLAG-MCM6 | This paper | N/A |
| pcDNA 3.0 FLAG-MCM7 | This paper | N/A |
| pcDNA 3.0 FLAG-MCM5 (D1) | This paper | N/A |
| pcDNA 3.0 FLAG-MCM5 (D2) | This paper | N/A |
| pcDNA 3.0 FLAG-MCM5 (D3) | This paper | N/A |
| pcDNA 3.0 FLAG-MCM5 (D4) | This paper | N/A |
| pcDNA 3.0 FLAG-MCM5 (D5) | This paper | N/A |
| pcDNA 3.0 FLAG-MCM5 (D6) | This paper | N/A |
| pcDNA 3.0 FLAG-MCM5 (D7) | This paper | N/A |
| pcDNA 3.0 FLAG-MCM5 (K583R) | This paper | N/A |
| pGEX-4T-1 MCM5 | This paper | N/A |
| pGEX-4T-1 MCM5 (K583R) | This paper | N/A |
| pcDNA 3.0 HA-UFM1 (delC2) | This paper | N/A |
| pcDNA 3.0 HA-UFM1 (R81C) | This paper | N/A |
| pcDNA 3.1 Myc-UBA5 | This paper | N/A |
| pcDNA 3.1 Myc-UBA5 (M57V) | This paper | N/A |

| Reagent/resource | Reference or source | Identifier or catalog number |
|---|---|---|
| pcDNA 3.1 Myc-UBA5 (V260M) | This paper | N/A |
| pcDNA 3.1 Myc-UBA5 (Q302Ter) | This paper | N/A |
| pcDNA 3.1 Myc-UBA5 (R55H) | This paper | N/A |
| pcDNA 3.1 Myc-UFC1 | This paper | N/A |
| pcDNA 3.1 Myc-UFC1 (T106I) | This paper | N/A |
| pcDNA 3.1 Myc-UFL1 | This paper | N/A |
| pcDNA 3.0 FLAG-UFL1 | This paper | N/A |
| pcDNA 3.1 Myc-UFBP1 | This paper | N/A |
| pcDNA 3.1 Myc-UFSP2 | This paper | N/A |
| pcDNA 3.1 Myc-UFSP2 (C302S) | This paper | N/A |
| pcDNA 3.0 FLAG-GINS3 | This paper | N/A |
| pcDNA 3.0 FLAG-CDC45 | This paper | N/A |
| pet28a-HA-UFM1 (delC2) | This paper | N/A |
| pet28a-UBA5 | This paper | N/A |
| pet28a-UFC1 | This paper | N/A |
| pet28a-UFL1 | This paper | N/A |
| pet28a-UFBP1 | This paper | N/A |
| **Antibodies** | | |
| Rabbit anti-MCM2 | Bethyl | Cat #A300-191A |
| Goat anti-MCM3 | Bethyl | Cat #A300-124A |
| Rabbit anti-MCM3 | CST | Cat #4012S |
| Rabbit anti-MCM4 | Bethyl | Cat #A300-193A |
| Rabbit anti-MCM5 | Bethyl | Cat #A300-195A |
| Rabbit anti-MCM5 | Proteintech | Cat #11703-1-AP |
| Rabbit anti-MCM7 | Bethyl | Cat #A302-585A |
| Rabbit anti-MCM7 | CST | Cat #3735S |
| Rabbit anti-MCM10 | Bethyl | Cat #A300-131A |
| Rabbit anti-PSF3 | Bethyl | Cat #A304-124A |
| Rabbit anti-PSF2 | Abcam | Cat #ab197123 |
| Rabbit anti-ORC1 | Bethyl | Cat #A301-892A |
| Rabbit anti-ORC2 | Bethyl | Cat #A302-735A |
| GINS4 | Abclonal | Cat #A8592 |
| Rabbit anti-PCNA | Abcam | Cat #ab92552 |
| Rabbit anti-H3 | Abcam | Cat # ab1791 |
| Rabbit anti-CDC45 | CST | Cat #11881S |
| Rabbit anti-CDC45 | Proteintech | Cat #15678-1-AP |
| Rabbit anti-UBA5 | Bethyl | Cat #A304-115A |
| Rabbit anti-UFM1 | Abcam | Cat #ab109305 |
| Rabbit anti-UFC1 | Proteintech | Cat #15783-1-AP |
| Rabbit anti-UFL1 | Bethyl | Cat #A303-455A |
| Rabbit anti-UFL1 | Bethyl | Cat #A303-456A |
| Rabbit anti-HA | Bethyl | Cat #A190-108A |
| Rabbit anti-HA | Proteintech | Cat #51064-2-AP |
| Mouse anti-FLAG | Sigma-Aldrich | Cat #F1804 |

| Reagent/resource | Reference or source | Identifier or catalog number |
|---|---|---|
| Rabbit anti-Actin | Abclonal | Cat #AC026 |
| Mouse anti-GST | MBL | Cat #M209-3 |
| Mouse anti-His | MBL | Cat #D291-3 |
| Rabbit anti-C-MYC | Bethyl | Cat #A190-105A |
| Rabbit anti-GAPDH | Bethyl | Cat #A300-641A |
| Mouse anti-Phospho-Histone H2A.X (Ser139) | CST | Cat #80312S |
| Rat anti-BrdU | Abcam | Cat #ab6326 |
| Mouse anti-BrdU | BD | Cat #347580 |
| Rabbit anti-IgG | Abclonal | Cat #AC005 |
| Donkey anti-rabbit IgG (H + L) | Jackson ImmunoResearch | Cat #711-035-152 |
| Goat anti-mouse IgG | Jackson ImmunoResearch | Cat #115-035-166 |
| Alexa Fluor® 488 Donkey anti-Rat IgG | Jackson ImmunoResearch | Cat #712-546-150 |
| Alexa Fluor 594 Donkey anti-mouse IgG | Life Technologies | Cat #1820027 |
| Rabbit anti-CHK1 | Bethyl | Cat #A300-298A |
| Rabbit anti-H2AX | Bethyl | Cat #A300-082A |
| Rabbit anti-UFBP1 | Proteintech | Cat #21445-1-AP |
| Rabbit anti-CDK5RAP3 | Proteintech | Cat #11007-1-AP |
| Rabbit anti-UFSP2 | Abcam | Cat #ab185965 |
| Rabbit anti-UBA5 | Proteintech | Cat #12093-1-AP |
| **Oligonucleotides and other sequence-based reagents** | | |
| qPCR primers | Wu et al, 2017 | Table EV1 |
| siRNA sequence | This study | Table EV1 |
| shRNA sequence | This study | Table EV1 |
| gRNA sequence | This study | Table EV1 |
| **Chemicals, enzymes, and other reagents** | | |
| 5-Chloro-2′-deoxyuridine | Sigma-Aldrich | Cat #C6891 |
| 5-Iodo-2′-deoxyuridine | Sigma-Aldrich | Cat #I7125 |
| 5-ethynyl-20-deoxyuridine | Invitrogen | Cat #A10044 |
| Biotin Azide | Invitrogen | Cat #B10184 |
| Copper (II) sulfate | Sigma-Aldrich | Cat #451657 |
| (+) Sodium L-ascorbate | Sigma-Aldrich | Cat #A4034 |
| DAPI solution (ready-to-use) | Solarbio | Cat #C0065 |
| DKM 2-93 | TargetMol® | Cat #T7415 |
| N-Ethylmaleimide | TargetMol® | Cat #T3088 |
| RO-3306 | TargetMol® | Cat # T2356 |
| XL413 | Selleck | Cat #S7547 |
| Aphidicolin | Sigma-Aldrich | Cat #38966-21-1 |
| Chelex 100 | Sigma-Aldrich | Cat #142-1253 |
| YF® 647A Click-iT EdU | UElandy | Cat #C6022L |
| Duolink In Situ Red Starter Kit | Sigma | Cat #DUO92101 |
| Anti-Flag Affinity Gel | Selleck | Cat #B23102 |
| Protein A SEPHAROSE | General Electric | Cat #17061801 |

| Reagent/resource | Reference or source | Identifier or catalog number |
|---|---|---|
| Lipofectamine-RNAiMAX | Thermo Scientific | Cat #13778150 |
| Glutathione-Sepharose 4B agarose | GE Healthcare | Cat #17075601 |
| HisSep Ni-NTA Agarose Resin 6FF | Yeasen | Cat #H6108050 |
| Polyethylenimine | Yeasen | Cat #MW40000 |
| Protein A/G Dynabeads | Millipore | Cat #17-10085 |
| ProteoSilver™ Silver Stain Kit | Sigma | Cat #PROTSIL1 |
| **Software** | | |
| Image J | NIH | https://imagej.nih.gov |
| FlowJo | Treestar | www.flowjo.com |
| GraphPad Prism 9 | GraphPad | www.graphpad.com |
| **Other** | | |

## Cell culture

Human HEK293T and HeLa cells were purchased from ATCC and cultured in Dulbecco's modified Eagle's medium (DMEM) supplemented with 10% fetal bovine serum (FBS) and 1% penicillin and streptomycin. The cells were maintained at 37 °C in a humidified incubator with an atmosphere containing 5% $CO_2$.

## Plasmids and transfection

Full-length cDNA clones encoding human MCM5 were amplified by PCR from cDNAs obtained from HeLa cells and then subcloned into a pcDNA3.0 3XFLAG vector (Invitrogen). UFM1 cDNA with two amino acids deleted from the C terminus (UFM1-delC2) and UFSP2 cDNA were cloned into a pcDNA3.0-HA vector. The cDNAs of UBA5, UFC1, UFL1, and UFBP1 were cloned into a pcDNA3.1-Myc vector. Point mutations in MCM5 (K583R) and UFSP2 (C302S) were generated using the Mut Express II Fast Mutagenesis Kit V2 (Vazyme). Bacteria BL21(DE3) cells expressing His-tagged UBA5, UFC1, UFM1, UFL1, and GST-tagged MCM5 were generated using the pET28a (Invitrogen) and pGEX-4T-1 (GE Healthcare) system. Plasmid transfections in HeLa or HEK293T cells were performed with polyethylenimine according to the manufacturer's conditions (YEASEN).

## Small interference RNAs

Endogenous UBA5, UFC1, UFL1, and MCM5 expression were knocked down using specific siRNAs purchased from Guangzhou RiboBio. The sequences were listed in Table EV1. The siRNAs were transfected into human cells using Lipofectamine RNAiMAX (Invitrogen), according to the manufacturer's conditions.

## UFL1 knockout (KO) cell line generation

UFL1 KO cell line was generated using CRISPR-Cas9 genome-editing technology. The sgRNA (CAACCGCCTAATCTCTTCCC)-containing PX459 (Addgene, 48139) plasmid was transfected into

HeLa cells. After 48 h, the cells were subcloned into 96-well plates after drug selection in the presence of 1 μg/ml puromycin for 24 h.

## Co-immunoprecipitation (Co-IP)

Immunoprecipitation in denaturing conditions (SDS-IP) was performed as previously described (Yoo et al, 2014). Briefly, harvested cells were lysed in lysis buffer (150 mM Tris, pH 8.0, 5% SDS, 30% glycerol) at 100 °C for 10 min and then digested in Benzonuclease (Sigma) at room temperature for 30 min. After centrifugation (12,000 × g, 10 min, 4 °C), the supernatant was diluted (20 times) with Buffer A [50 mM Tris-HCl (pH 8), 150 mM NaCl, 1% Triton X-100, 1× protease inhibitor cocktail (PIC), and 2 mM N-Ethylmaleimide] and immunoprecipitated with the relevant antibodies.

## Western blotting (WB)

Samples were separated by SDS-PAGE (Bio-Rad) and transferred to PVDF membranes (Cytiva). After skim milk (Solarbio) blocking, the membranes were incubated with primary antibody at 4 °C overnight. The membranes were incubated with secondary antibodies for 1 h at room temperature, then visualized using electro-chemiluminescence detection reagents (SuperSignal™ West Pico PLUS, Thermo).

## Chromatin fractionation (CF)

CF was performed using two different methods as previously described (Wu et al, 2022; Mendez and Stillman, 2000; Kannouche et al, 2004). The method used in Fig. S4 corresponds to the approach described before (Mendez and Stillman, 2000), and the second method which was used for all other chromatin fractionation experiments in this paper Mol Cell 2004 (Kannouche et al, 2004), Briefly, cells were harvested by trypsinization and washed with ice-cold PBS. Then, the cells were lysed with CSK 100 buffer [100 mM NaCl, 300 mM sucrose, 3 mM $MgCl_2$, 10 mM Pipes (pH 6.8), 1 mM EGTA, 0.2% Triton X-100] containing 1× protease inhibitor cocktail (PIC) for 5 min on ice. After centrifugation (1200 × g, 10 min), the supernatant was removed; the pellet was washed with ice-cold PBS and resuspended in sample buffer before analysis by WB.

## Silver staining

HEK293T cells were lysed and immunoprecipitated with an anti-FLAG antibody. The immunoprecipitates were resuspended in a sample buffer and loaded onto an SDS gel. The gel was then stained according to the manufacturer's instructions (ProteoSilver™ Plus, Sigma).

## Mass spectrometric analysis

HEK293T cells were lysed under non-denaturing (for protein–protein interaction identification) or denaturing (for UFMylation substrate screening) conditions and immunoprecipitated with a FLAG antibody. The immunoprecipitates were then analyzed by mass spectrometry externally (Wininnovate Bio).

## Double thymidine block (dT)

HeLa cells were treated with thymidine (2 mM) for 18 h, followed by three washes with PBS, and then cultured in fresh medium for 8 h. Thymidine was then added back into the medium to a final concentration of 2 mM for another 18 h. The cells were washed a further three times with PBS and then cultured in fresh medium for the indicated time before being harvested.

## Cell growth assay

Cell growth was determined by counting viable cells each day for 4 days after seeding. The number of cells was then compared to day zero, and the data was expressed as the means of three independent experiments.

## Isolation of proteins on nascent DNA (iPOND) assay

An iPOND assay was performed as previously described (Sirbu et al, 2012). Briefly, growing HeLa cells were labeled with 10 μM EdU for 10 min. The cells were then subjected to cross-linking with 1% formaldehyde for 20 min and then quenched with 0.125 M glycine for 5 min, all at room temperature (RT). The cell pellets were collected and washed with ice-cold PBS three times and then resuspended in 0.25% Triton X-100 in PBS and incubated for 30 min on ice for permeabilization. Next, the cells were washed once with ice-cold 0.5% BSA/PBS and once with PBS before being incubated with click reaction buffer (2 mM CuSO4, 10 μM biotin azide, and 10 mM sodium ascorbate) for 2 h. Following the click reaction, the cells were washed once with ice-cold 0.5% BSA/PBS and once with PBS before being resuspended in lysis buffer (1% SDS, 50 mM Tris [pH 8.0]). After sonicating (30 s-on, 30 s-off, 10 cycles) in a bioruptor (Diagenode) at 4 °C, the cell lysates were centrifuged and the supernatant was diluted 1:1 with ice-cold PBS containing a protease inhibitor cocktail (PIC). Streptavidin agarose resin M280 (Millipore) was incubated with the samples and rotated at 4 °C overnight. The captured proteins were washed twice with lysis buffer, then once with 1 M NaCl before being eluted by incubation in 6x sample buffer at 95 °C for 25 min.

## DNA fiber assay

DNA fiber assays were performed as previously described (Nieminuszczy et al, 2016). Briefly, HeLa cells were sequentially labeled with 40 μM CldU and 100 μM IdU for 30 min each at 37 °C in a humidified incubator with a 5% CO$_2$ atmosphere. After labeling, the cells were immediately harvested by trypsinization, washed three times with ice-cold PBS, and then diluted to $3 \times 10^5$ cells/ml. Before spreading, the labeled cells were mixed with unlabeled cells at a ratio of 1:2. Then, 2.5 μl of the cell suspension was loaded onto a microscope slide and air dried for 3 min at RT. Subsequently, 10 μl lysis solution (200 mM Tris-HCl, pH 7.5, 50 mM EDTA, and 0.5% SDS) was added to the cell suspension, mixed by gentle swirling with a pipette tip, and incubated again at RT for 5 min before spreading. Following spreading, the slides were fixed in methanol/acetic acid (3:1) solution at 4 °C for 1 h and then denatured in 2.5 N HCl for 90 min. After three washes with PBS, the slides were blocked in 1% BSA (in PBST) for 30 min and

incubated with a rat anti-BrdU antibody (Abcam BU1/75, 1:250) and anti-BrdU (BD B44, 1:50) in blocking buffer at 4 °C overnight. After three washes with PBST, the slides were incubated with Alexa Fluor 488-conjugated anti-rat (Jackson ImmunoResearch, 1:200) and Alexa Fluor 594-conjugated anti-mouse antibodies at 37 °C for 45 min in the dark. After three washes in PBST, the slides were mounted in an antifade before images were captured under a fluorescence microscope (OLYMPUS, CKX53) using a 60x objective.

Replication origins are identified by red–green–red fiber patterns, where a green segment (CldU-labeled) is flanked by red segments (IdU-labeled) on both sides. Occasionally, the green segment may appear split by an unstained region; this reflects bidirectional fork movement away from the origin during CldU incorporation. Only fibers lying exactly on a horizontal line are considered to be from the same DNA sample and are counted. The midpoint between two adjacent green segments is taken as the origin location. The IOD is then calculated by measuring the linear distance between these origins along the same DNA fiber, with values expressed in micrometers to reflect the physical spacing between replication origins during DNA synthesis.

## ChIP assay

Asynchronous HeLa cells were harvested and fixed with 1% formaldehyde for 20 min, then quenched with 0.125 M glycine for 5 min, all at RT. After centrifugation, the cell pellets were resuspended in sonication buffer [1% SDS, 10 mM EDTA, 50 mM Tris (pH 8.0)] containing protease inhibitors and incubated for 10 min on ice. The cells were then sonicated using a Bioruptor at 4 °C (2 × 10 cycles of 30 s-on and 30 s-off) to obtain 500 bp fragments. The sheared chromatin was cleared by centrifugation before the supernatant was incubated with MCM3, MCM7, or FLAG-recognizing antibodies overnight at 4 °C. After immunoprecipitation, Protein A/G Dynabeads (Millipore 17-10085) were added for 1 h and then washed sequentially with low salt buffer (0.1% SDS, 1% Trition X-100, 2 mM EDTA, 20 mM Tris-HCl, pH 8.0, 150 mM NaCl), high salt buffer (0.1% SDS, 1%Trition X-100, 2 mM EDTA, 20 mM Tris-HCl, pH 8.0, 500 mM NaCl), LiCl-containing buffer (0.25 M LiCl, 1.0% NP-40, 1.0% deoxycholic acid, 1.0 mM EDTA, 10 mM Tris-HCl, pH 8.1) and TE buffer (10 mM EDTA, 10 mM Tris-HCl, pH 8.0), all at RT. After washing, the beads were isolated in elution buffer (1% SDS, 0.1 M NaHCO$_3$) for 15 min by gentle vortexing of the samples. The protein/DNA complexes were eluted and subjected to reverse cross-linking by mixing with 4 μl 5 M NaCl and proteinase K at 65 °C for 2 h. Finally, the DNA was eluted in Chelex 100 solution and prepared for qPCR.

The primers used for qPCR were as previously described (Wu et al, 2017). The sequences were listed in Table EV1.

## EdU incorporation assay

An EdU incorporation assay was completed according to the instructions provided by the manufacturer of the YF® 647A Click-iT EdU Imaging Kit (UElandy, C6046L). Briefly, cells were pulsed with 10 μM EdU for 20 min at 37 °C in a humidified incubator with a 5% CO$_2$ atmosphere, then fixed in 4% paraformaldehyde (PFA)

for 10 min, quenched with 0.125 M glycine for 5 min and then permeabilized with 0.5% Triton X-100 in PBS for 20 min, all at RT. The cells were click-reacted and stained according to the manufacturer's instructions (YF® 647A Click-iT EdU Imaging Kit). ImageJ software (Version 1.53a, https://imagej.net) was used to quantify the fluorescence intensity.

## Protein purification

GST-tagged MCM5 (WT and K583R), His-tagged UFMylation enzymes (UBA5, UFC1, UFL1, UFBP1), and His-HA-tagged UFM1 (delC2) were expressed in BL21 cells, which were then sonicated (120 W, 30 s-on and 30 s-off, 10 cycles). The tagged proteins were purified using Glutathione-Sepharose 4B agarose (GE Healthcare) and Ni-NTA agarose (Yeasen).

## In vitro UFMylation assay

Bacterially produced His-UBA5 (0.1 μM), His-UFC1 (0.1 μM), His-UFL1 (0.1 μM), His-HA-UFM1ΔC2 (0.1 μM) and GST-MCM5 (0.1 μM) were mixed in reaction buffer (0.05% BSA, 50 mM HEPES, pH 7.5) containing 5 mM γ-ATP and 10 mM $MgCl_2$, and incubated at 30 °C for 90 min. The mixtures were then diluted with ice-cold PBS and added to GST agarose resin (Cytiva) for 1 h at 4 °C. After washing with PBS three times, the beads were boiled in 6× SDS sample buffer for 10 min.

## Proximity ligation assay (PLA)

Cells were fixed in 4% PFA at RT for 10 min and then permeabilized with 0.5% Triton X-100 for 5 min. After blocking with 2% BSA for 1 h, the cells were incubated with primary antibodies at 4 °C overnight. PLA was then carried out using the Duolink In Situ Red Starter kit (Sigma-Aldrich), according to the manufacturer's instructions.

## Mitotic spread analysis

HeLa cells depleted of UFL1 or MCM5 were treated with 10 μM nocodazole for 6 h before harvesting. The cells were then treated with 56 mM KCl at 37 °C for 20 min and fixed in methanol/acetic acid (3:1) at RT. Chromosome spreads were prepared on slides and stained with Giemsa for 10 min at RT. Images were captured using a DragonFly confocal imaging system (Andor).

## Flow cytometry

HeLa cells were fixed in 4% PFA for 10 min at RT and then permeabilized with 0.5% Triton X-100 for 20 min. After three washes with PBS, the cells were stained with propidium iodide (PI) at 37 °C for 40 min or with DAPI at RT for 5 min. The cell cycle profile was then analyzed using a flow cytometer (Cytoflex, Beckman).

## Statistical analysis

The data are presented as the means ± standard deviation (SD) from three independent experiments and analyzed using GraphPad Prism software (Version 9.4.1). A two-tailed unpaired Student's $t$ test with Welch's correction was used to determine statistical significance between the two groups. Data from multiple groups were analyzed using Ordinary one-way ANOVA, followed by a Sidak's test. Multiple groups with two variables were analyzed using two-way ANOVA, followed by Sidak's test. $P < 0.05$ was considered statistically significant.

## Data availability

This study includes no data deposited in external repositories.

The source data of this paper are collected in the following database record: biostudies:S-SCDT-10_1038-S44318-025-00562-6.

## Peer review information

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

## Acknowledgements

The authors would like to thank the members of the Xu lab for their insightful discussions and assistance. This work was supported by the National Natural Science Foundation of China (grant U24A20740, 32090031, 82072947, 32250710138), the National Key R&D Program of China (2022YFA1302804), the Pearl River Talent Recruitment Program (2021JC02Y089), Shenzhen Medical Research Fund (D2403014), and the Shenzhen Science and Technology Innovation Commission (grant JCYJ20220818095616035, JCYJ20220818095605011). We would like to thank Dr. Hiroyuki Araki (National Genetic Institute, Mishima, Japan) for critical reading.

## Author contributions

**Zheng Li**: Validation; Investigation; Methodology; Writing—original draft. **Xingxuan Wu**: Investigation; Methodology. **Liu Liu**: Formal analysis; Investigation. **Shaohong Rao**: Resources. **Yanting Liao**: Resources. **Mengting Liu**: Resources. **Bin Peng**: Formal analysis. **Qiongdan Zhang**: Formal analysis. **Yisui Xia**: Formal analysis; Writing—review and editing. **Yuanliang Zhai**: Validation; Project administration; Writing—review and editing. **Shunichi Takeda**: Supervision; Project administration; Writing—review and editing. **Xingzhi Xu**: Supervision; Funding acquisition; Project administration; Writing—review and editing.

Source data underlying figure panels in this paper may have individual authorship assigned. Where available, figure panel/source data authorship is listed in the following database record: biostudies:S-SCDT-10_1038-S44318-025-00562-6.

## Disclosure and competing interests statement

The authors declare no competing interests.

# Expanded View Figures

**Figure EV1.   UFMylation is required for DNA replication.**

(**A**) Western blot analysis of protein expression in HeLa cells transfected with the indicated siRNAs and over-expression constructs. (**B**) Growth curves of the HeLa cells in (**A**). The mean number of cells (biological replicates, $n = 3$) ± SD is shown. *P* values were calculated by Two-way ANOVA (***$P < 0.001$; ****$P < 0.0001$). *P* values: siNC vs siUFL1, day 2/3/4, 5.33e-004/1.86e-009/9.67e-014. (**C**) Western blot analysis of UFL1 expression in *WT* or *UFL1* knockout (UFL1 KO) HeLa cells. (**D**) Growth curves of the indicated HeLa cells as in (**C**). The mean number of cells (biological replicates, $n = 3$) ± SD is shown. *P* values were calculated by Two-way ANOVA (****$P < 0.001$). *P* values: WT vs UFL1 KO, day 2/3/4, 8.97e-007/4.8e-012/2.71e-016. (**E**) Growth curves of the indicated HeLa cells. The mean number of cells (biological replicates, $n = 3$) ± SD is shown. *P* values were calculated by Two-way ANOVA (****$P < 0.001$; ns $P > 0.05$, no significance). *P* values: WT vs WT + DKM2-93, day 4, 1.66e-013; UFL1 KO vs UFL1 KO + DKM 2-93, day 4, 0.713. (**F, G**) Quantification of the EdU MFI in U2OS (**F**) and A549 (**G**) cells pre-treated with DMSO or DKM 2-93 (100 μM). Data are presented as mean ± SD of three biological replicates ($n = 3$). *P* value was calculated by unpaired *t* test with Welch's correction (***$P < 0.001$; ****$P < 0.0001$). *P* value: DMSO vs DKM 2-93 in (**E**)/(**F**), 2.77e-007/1.35e-004. (**H**) Quantification of the ratio of red-green-red (RGR) tracks to the total number of red-green (RG) fork tracks. Data are presented as mean ± SD. *P* value was calculated by unpaired *t* test with Welch's correction (****$P < 0.0001$). *P* value, 4.20352928e-005. (**I**) Quantification of the IdU track length in U2OS (Left) and A549 (Right) cells transfected with the indicated siRNAs and plasmids and sequentially labeled with CldU and IdU for 30 min. Data are presented as in Fig. 1E. Data are presented as mean ± SD. *n* DNA fiber number. *P* values were calculated by Ordinary one-way ANOVA (****$P < 0.0001$). *P* values: siNC vs siUFL1 in (**H**)/(**I**), 1.78e-025/1.89e-019; siUFL1 vs siUFL1+HA-UFL1 in (**H**)/(**I**), 4.52e-018/3.11e-013. (**J**) Western blot analysis of protein expression in HeLa cells transfected with the indicated siRNAs. (**K, L**) Quantification of the IdU track length in U2OS (**K**) and A549 (**L**) cells pre-treated with DKM 2-93 (100 μM) for 1 h before CldU and IdU labeling in the presence of DKM 2-93 for 30 min. Data are presented as mean ± SD. *n* DNA fiber number. *P* values were calculated by unpaired *t* test with Welch's correction (****$P < 0.0001$). *P* values: DMSO vs DKM 2-93 in (**K**)/(**L**), 9.04e-012/9.31e-016. (**M**) Western blot analysis of protein expression in HeLa cells treated with DMSO, DKM 2-93 (100 μM), hydroxyurea (2 mM), or bleomycin (10 μg/ml) for 1 h. Asterisk represents non-specific band. (**N–P**) Western blot analysis of protein expression in indicated HeLa cells used in Fig. 1G.

                                                            

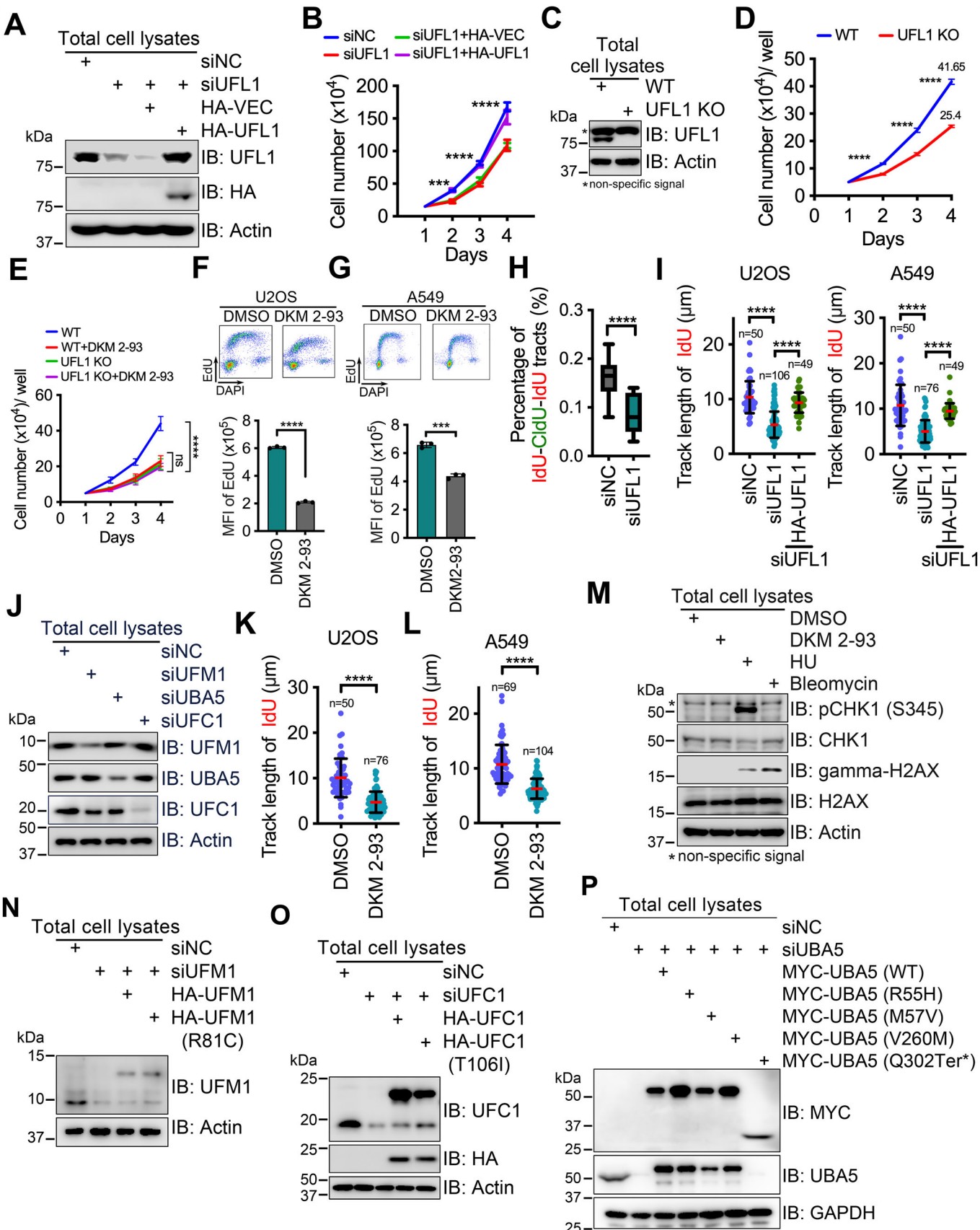

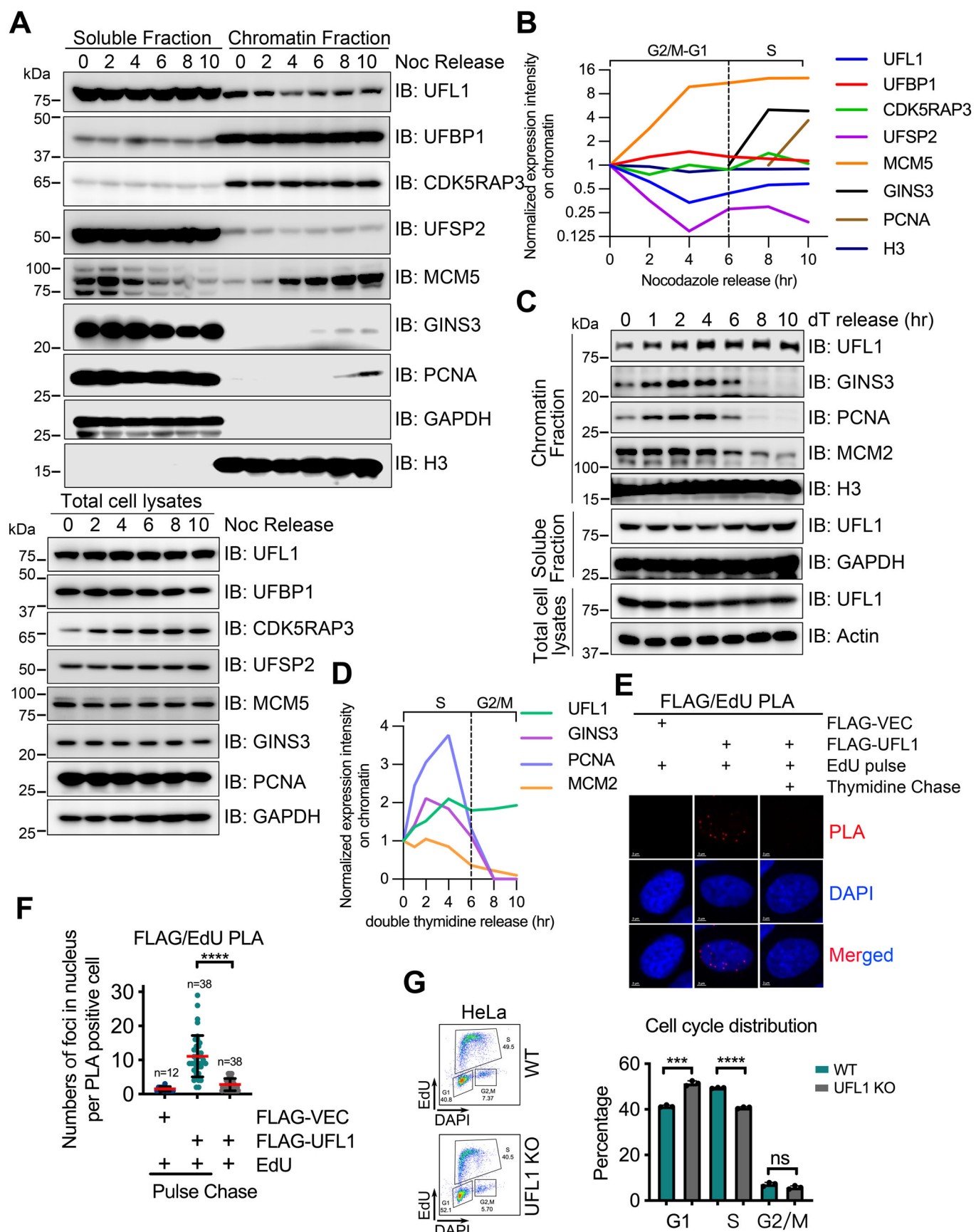

◀ **Figure EV2. UFL1 is present at replication origins and affects their timing.**

(A–D) The protein level of UFL1 and replication factors in the soluble fraction, chromatin fraction and total cell lysates of HeLa cells synchronized with nocodazole (A, B) and dT (C, D) block. At the time zero, cells were released to the cell cycle. Western blot analysis of proteins is presented in (A, C), and the quantification of analyzed proteins normalized to H3 is presented in (B, D). (E, F) Quantification of PLA foci in HeLa cells transiently transfected with FLAG-VEC or FLAG-UFL1. Cells were labeled with EdU for 30 min. After labeling, the chase-group cells were additionally labeled with thymidine for 30 min. Representative images of PLA foci (red) (E) and the average number of foci per nucleus (F) are shown. $n$ cell number. $P$ values was calculated by unpaired $t$ test with Welch's correction (****$P < 0.0001$). $P$ value: pulse vs chase in FLAG-UFL1 expressed group, 3.22e-010. Scale bar, 3 µm. (G) Cell cycle analysis of *WT* or UFL1 KO HeLa cells. The *X*-axis is the intensity of DAPI staining on a linear scale, and the *Y*-axis is EdU uptake on a log scale. *WT* or UFL1 KO HeLa cells were pulse-labeled with EdU for 30 min. Representative data and quantifications are shown in the left and right panels, respectively. Data are presented as mean ± SD of three biological replicates ($n = 3$). $P$ value was calculated by Ordinary one-way ANOVA (***$P < 0.001$; ****$P < 0.0001$; ns $P > 0.05$, no significance). $P$ value: WT vs UFL1 KO in G1 phase/S phase/G2/M phase, 1.27e-009/6.93e-009/0.0691.

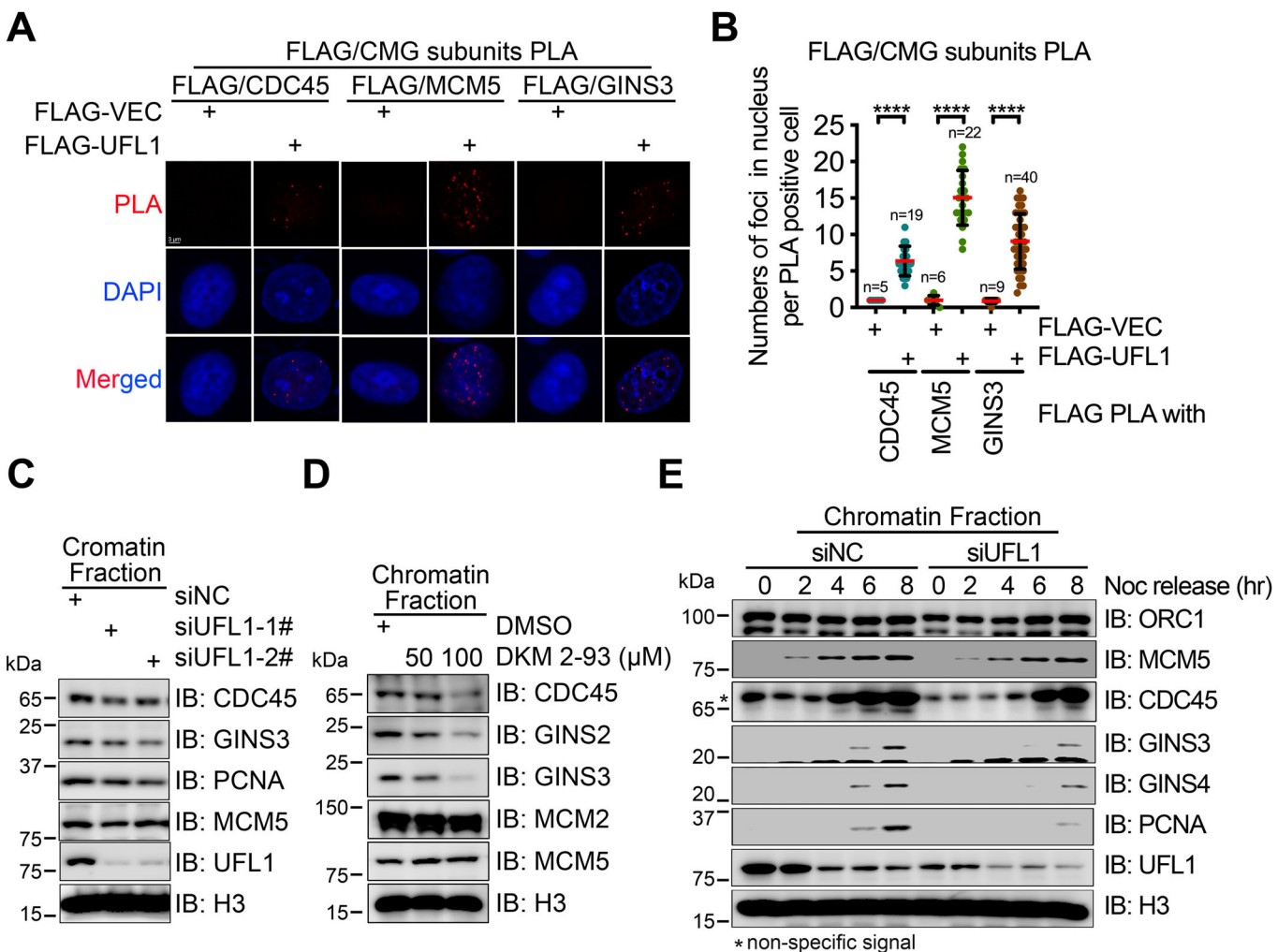

**Figure EV3. UFL1 binds to CMG helicase components and affects their loading to replisome.**

(A, B) PLA assays to analyze the proximal localization of UFL1 by CDC45, GINS3, and MCM5 in HeLa cells transfected with FLAG-VEC or FLAG-UFL1. Representative images of PLA foci (red) (A) and the average number of foci per nucleus per PLA positive cell (B) are shown. The scale bar is 3 μm. Data are presented as mean ± SD. *n* cell number. *P* values was calculated by unpaired *t* test with Welch's correction (****$P < 0.0001$). *P* value: FLAG-VEC vs FLAG-UFL1 in CDC45/MCM5/GINS3 group, 9.87e-010/6.07e-015/1.08e-016. Scale bar, 3 μm. (C) Western blot analysis of chromatin fractions of HeLa cells that were transfected with a non-targeting control (siNC) or two siRNA oligos targeting UFL1 (siUFL1-1#, siUFL1-2#). (D) Western blot analysis of chromatin fractions of HeLa cells that were treated with DMSO or DKM 2-93 (100 μM, 16 h). (E) Western blot analysis of chromatin fractions of HeLa cells that were transfected with siNC or siUFL1 at the indicated time points after release from nocodazole block.

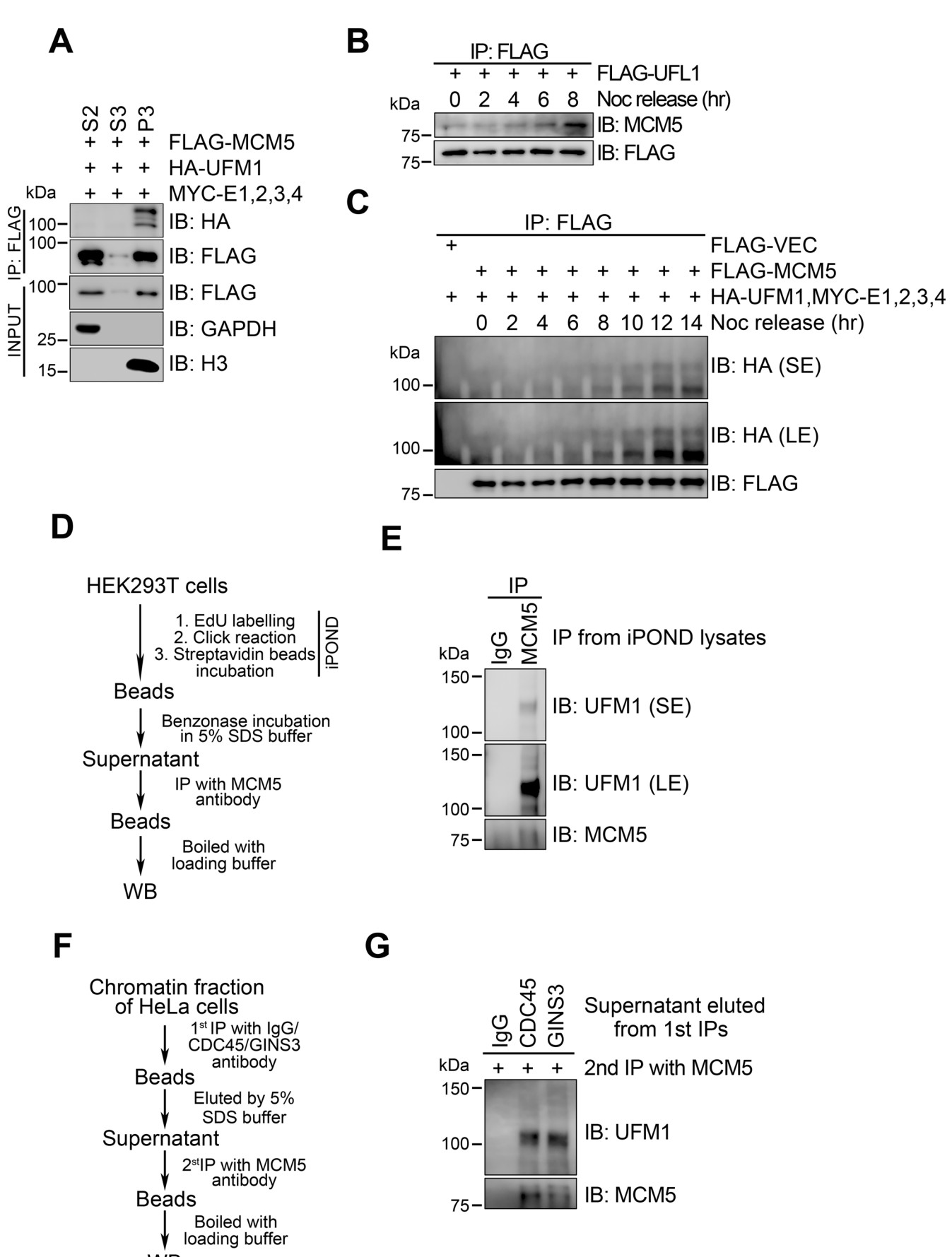

**Figure EV4.   The UFMylation of MCM5 occurs at the ongoing replication fork.**

(A) The cytoplasm (S2), nucleoplasm (S3), and chromatin (P3) fractions of HeLa cells expressing the indicated plasmids were subjected to IP with a FLAG antibody and western blotting with the indicated antibodies. (B) HeLa cells expressing FLAG-UFL1 were synchronized with nocodazole (as in Fig. 3F) and then lysed at the indicated times. UFL1 was purified on FLAG M2 beads and analyzed by western blotting. (C) HeLa cells expressing UFMylation factors were synchronized as in (B). After release from the nocodazole block, the cells were lysed at the indicated times. MCM5 was purified on FLAG M2 beads and subjected to western blotting with the indicated antibodies. SE short exposure, LE long exposure. (D) Schematic of the combined IP and iPOND assay. (E) HEK293T cell lysates were purified as in (D) and subjected to western blotting with the indicated antibodies. (F) Schematic of the two-step IP. (G) HeLa cell lysates were purified as in (F) and subjected to western blotting with the indicated antibodies.

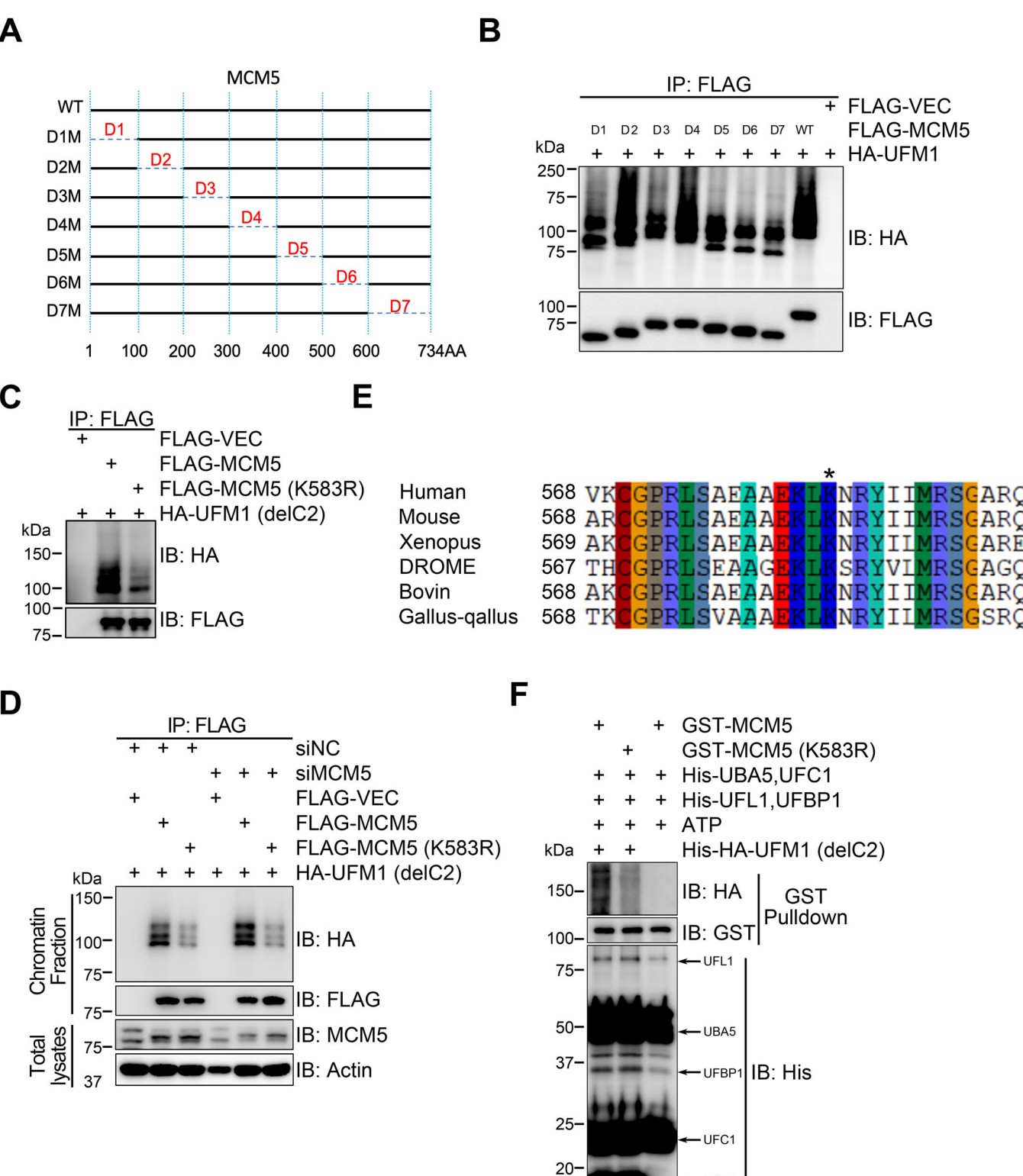

◀ **Figure EV5.　MCM5 is UFMylated on Lys583.**

(A) The structure of MCM5 and the various deletions D1–D7. (B) HEK293T cells were transfected with the indicated plasmids, and the lysates were immunoprecipitated with a FLAG antibody before western blotting with the indicated antibodies. (C) HEK293T cells were transfected with the indicated plasmids and treated as in (B). (D) HeLa cells were transfected with the indicated plasmids, and the chromatin fraction were isolated for subsequent immunoprecipitation and western blotting as in (B). (E) Sequence alignment of the region encompassing the UFMylation site of MCM5 from indicated species. The asterisk shows the position of the K583. (F) Bacterially produced UFMylation factors [His-UBA5, His-UFC1, His-UFL1, and His-HA-UFM (delC2)] and bacterially produced GST-MCM5 or GST-MCM5 (K583R) were incubated with GST beads before analysis by western blotting with the indicated antibodies.

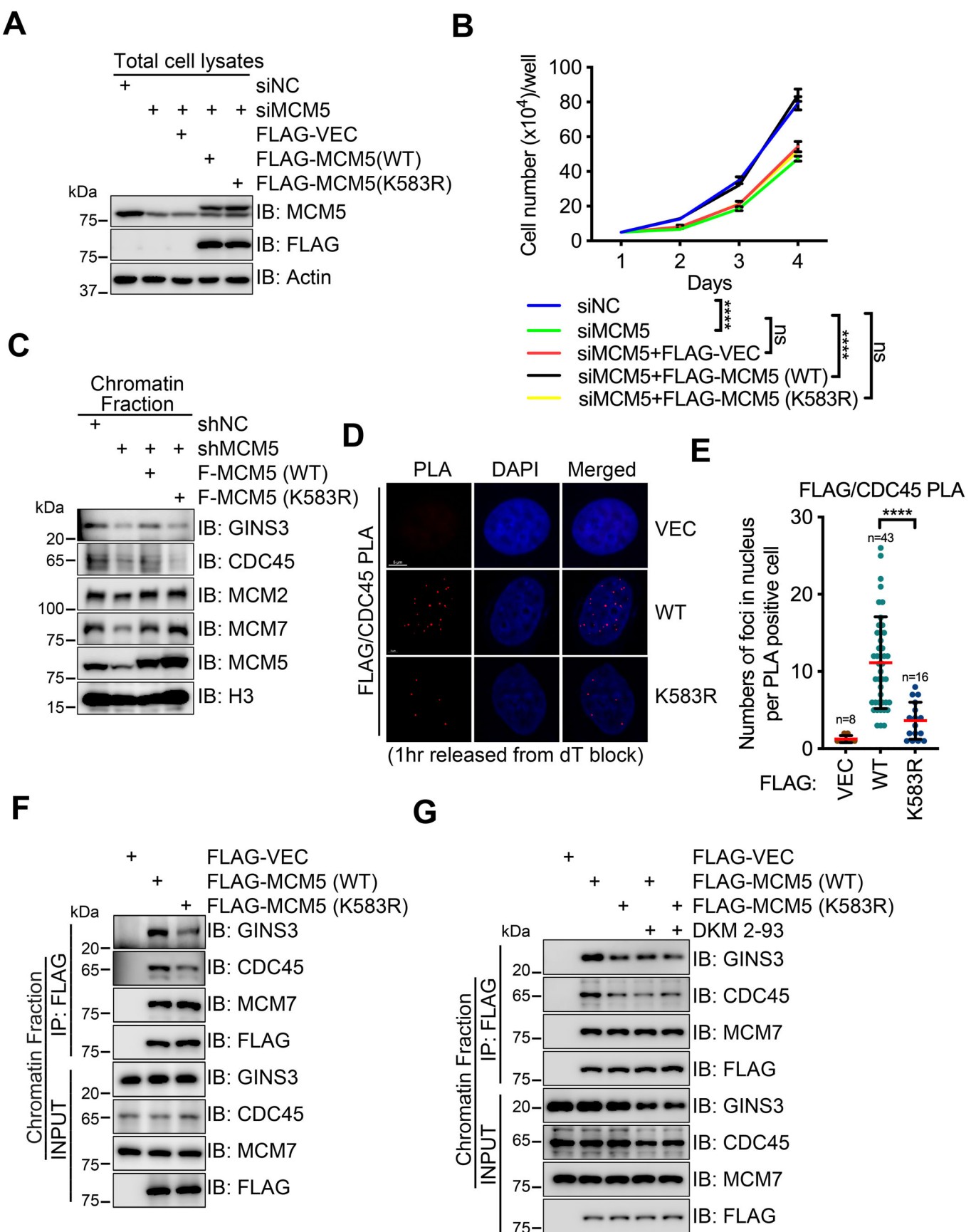

**Figure EV6.   MCM5 UFMylation at K583 stabilizes CMG helicase complex formation.**

(A) Western blot analysis of the indicated proteins in HeLa cells transfected with indicated siRNAs and plasmids. (B) Growth curves of the HeLa cells used in (A). The mean number of cells (biological replicates, $n = 3$) ± SD is shown. $P$ values were calculated by Tukey's multiple comparisons test (****$P < 0.0001$; ns $P > 0.05$, no significance). $P$ values: siNC vs siMCM5, 3.17e-005; siMCM5 vs siMCM5+FLAG-VEC, 0.99; siMCM5 vs siMCM5+FLAG-MCM5 (WT), 1.58e-005; siMCM5 vs siMCM5+FLAG-MCM5 (K583R), 0.93. (C) Western blot analysis of the proteins in chromatin fractions of HeLa cells transiently transfected with the indicated shRNA and plasmids. (D, E) Immunostaining of PLA foci in HeLa cells transiently transfected with either FLAG-VEC, FLAG-MCM5 (WT), or FLAG-MCM5 (K583R) at 1 h released from the dT block. Representative PLA foci (red) images are shown in (D), and the quantitative data are shown in (E). $n$ cell number. $P$ values was calculated by unpaired $t$ test with Welch's correction (****$P < 0.0001$). $P$ value: WT vs K583R, 4.62e-009. Scale bar, 5 μm. (F, G) Western blot analysis of proteins in anti-FLAG immunoprecipitants enriched from the chromatin fraction of HeLa cells transfected with indicated plasmids. The HeLa cells in (G) were treated with DKM 2-93 (100 μM) for 16 h before IP of the chromatin fraction.

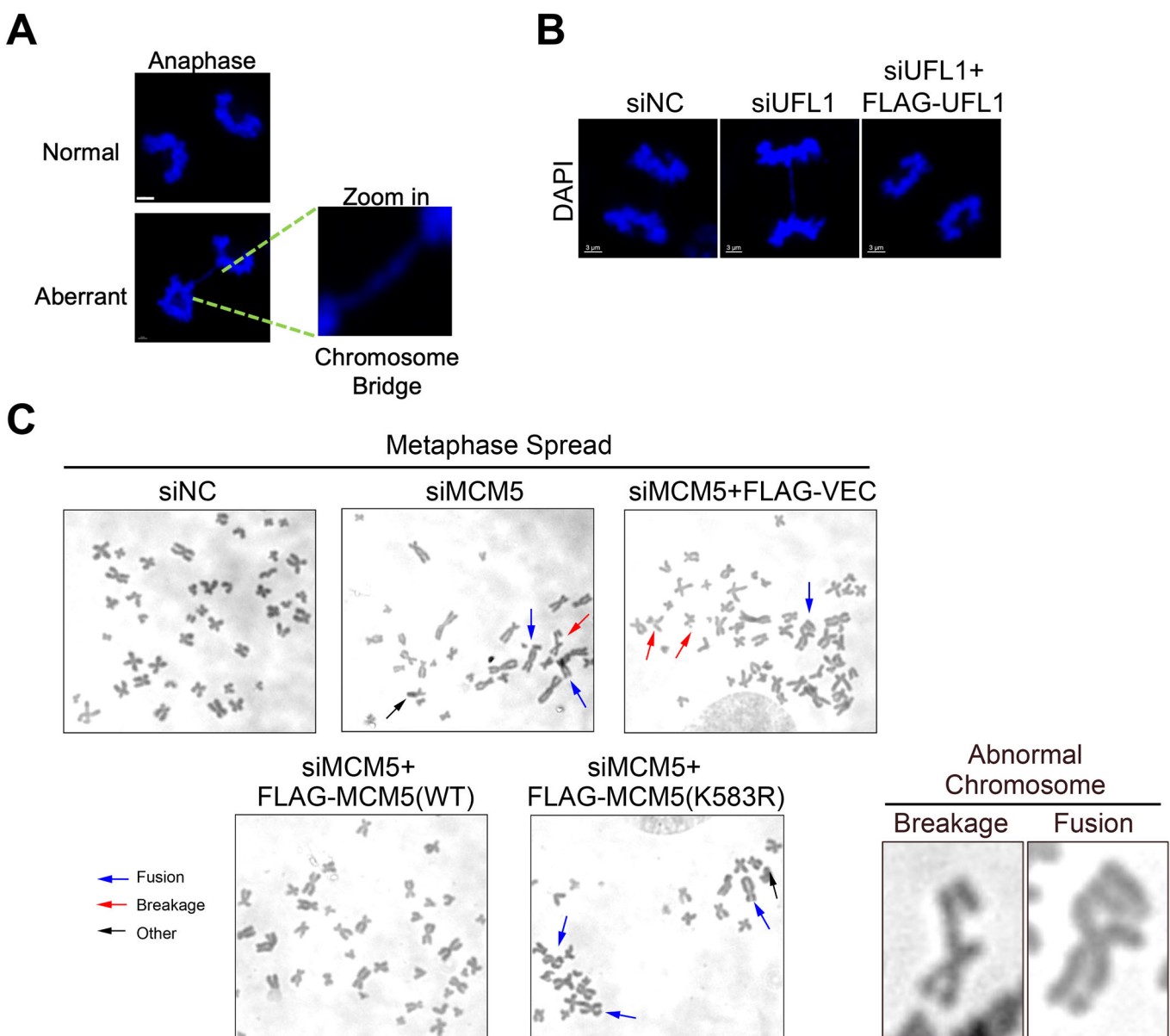

**Figure EV7.   MCM5 UFMylation promotes cell growth and maintains genome stability.**

(A, B) Representative images of chromosome bridge events in HeLa cells during cell cycle anaphase. The Hela cells in (A) were fixed and permeabilized before staining with DAPI (blue). The HeLa cells in (B) were transfected with the indicated siRNAs and FLAG-UFL1 and were treated as in (A). Scale bar, 3 μm. (C) Representative images of chromosomes obtained from HeLa cells transfected with the indicated siRNA oligos and plasmids. The chromosomes were stained with Giemsa before images were captured. The abnormal chromosomes are shown at a higher magnification in the right panel.

