## [Peer Review File · The EMBO Journal]

MCM5 UFMylation regulates replication origin firing and fork progression

Zheng Li, Xingxuan Wu, Liu Liu, Shaohong Rao, Yanting Liao, Mengting Liu, Bin Peng, Qiongdan Zhang, Yisui Xia, Yuanliang Zhai, Shunichi Takeda, and Xingzhi Xu

Corresponding author(s): Xingzhi Xu (Xingzhi.Xu@szu.edu.cn) , Xingzhi Xu (Xingzhi.Xu@szu.edu.cn), Shunichi Takeda (stakeda@szu.edu.cn)

Review Timeline:

Submission Date:	4th Dec 24
Editorial Decision:	7th Feb 25
Revision Received:	6th May 25
Editorial Decision:	5th Jun 25
Revision Received:	8th Aug 25
Accepted:	27th Aug 25

Editor: Hartmut Vodermaier

Transaction Report:

Prof. Shunichi Takeda
Shenzhen University
China

7th Feb 2025

Re: EMBOJ-2024-119820
MCM5 UFMylation regulates replication origin firing and fork progression

Dear Drs. Takeda and Xu,

Thank you again for submitting your study on MCM5 UFMylation for our consideration. I apologize for the delay in getting back to you with a decision, mainly due to a large submission backlog and limited reviewer availability during December and January. We have now finally received the feedback from three experts in UFMylation (referee 1) and DNA replication control (referees 2 & 3), copied below for your information. As you will see, all referees appreciate the interest and potential importance of your findings. Nevertheless, they also raise a number of substantive concerns that would need to be addressed prior to publication, including the major question of physiological occurrence of MCM5 UFMylation.

Should you be able to satisfactorily address the referees' major points, we would be open to considering a revised version further for publication. However, I have to make clear that we would only return it to the referees if we felt that the key issue of endogenous MCM5 UFMylation had been decisively demonstrated. Furthermore, I should stress that our single-major-revision-round policy makes it important to diligently respond to all points raised by the reviewers at the time of resubmission.

In this light, I would very much encourage you to contact me with a revision plan and preliminary point-by-point response already during the early stages of your revision work, so that we could discuss if and how the main points could be resolved, or if a less completely revised manuscript might alternatively become suitable for publication in one of our sister journals like EMBO Reports or Life Science Alliance. Of course, we would also be open to extension of the default three-months revision period if needed; our 'scooping protection' (meaning that competing work appearing elsewhere in the meantime will not affect our considerations of your study) would of course remain valid also throughout such an extension.

Detailed information on preparing, formatting and uploading a revised manuscript can be found below and in our Guide to Authors. Thank you again for the opportunity to consider this work for The EMBO Journal, and I look forward to hearing from you in due time.

With kind regards,

Hartmut

- 3) Revised manuscript text (including main tables, and figure legends for main and EV figures) has to be submitted as editable text file (e.g., .docx format). We encourage highlighting of changes (e.g., via text color) for the referees' reference.
- 4) Each main and each Expanded View (EV) figure should be uploaded as individual production-quality files (preferably in .eps, .tif, .jpg formats). For suggestions on figure preparation/layout, please refer to our Figure Preparation Guidelines: <http://bit.ly/EMBOPressFigurePreparationGuideline>
- 5) Point-by-point response letters should include the original referee comments in full together with your detailed responses to them (and to specific editor requests if applicable), and also be uploaded as editable (e.g., .docx) text files.
- 6) Please complete our Author Checklist, and make sure that information entered into the checklist is also reflected in the manuscript; the checklist will be available to readers as part of the Review Process File. A download link is found at the top of our Guide to Authors: embopress.org/page/journal/14602075/authorguide
- 7) All authors listed as (co-)corresponding need to deposit, in their respective author profiles in our submission system, a unique ORCID identifier linked to their name. Please see our Guide to Authors for detailed instructions.
- 8) Please note that supplementary information at EMBO Press has been superseded by the 'Expanded View' for inclusion of additional figures, tables, movies or datasets; with up to five EV Figures being typeset and directly accessible in the HTML version of the article. For details and guidance, please refer to: embopress.org/page/journal/14602075/authorguide#expandedview
- 9) To facilitate reproducibility and cross-laboratory adoption of methodologies, please structure the Materials & Methods section as outlined in our guide to authors, including a completed Reagents and Tools Table that can be downloaded from our author guidelines as well (<https://www.embopress.org/page/journal/14602075/authorguide#structuredmethods>).
- 10) Digital image enhancement is acceptable practice, as long as it accurately represents the original data and conforms to community standards. If a figure has been subjected to significant electronic manipulation, this must be clearly noted in the figure legend and/or the 'Materials and Methods' section. The editors reserve the right to request original versions of figures and the original images that were used to assemble the figure. Finally, we generally encourage uploading of numerical as well as gel/blot image source data; for details see: embopress.org/page/journal/14602075/authorguide#sourcedata

At EMBO Press, we ask authors to provide source data for the main manuscript figures. Our source data coordinator will contact you to discuss which figure panels we would need source data for and will also provide you with helpful tips on how to upload and organize the files.

Revision to The EMBO Journal should be submitted online within 90 days, unless an extension has been requested and approved by the editor; please click on the link below to submit the revision online before 8th May 2025:
Link Not Available

If you choose to alternatively have this study further considered by another EMBO Press publication, please use the following hyperlink to directly transfer the manuscript, optionally with inclusion of referee reports and identities:
Link Not Available

Referee #1:

General Comments

This manuscript explores the role of UFMylation, specifically MCM5 UFMylation at Lys583, in DNA replication, with a focus on its impact on replication origin firing and fork progression. The study provides important insights into the mechanistic link between UFMylation and the CMG helicase complex, highlighting its relevance in diseases such as microcephalic primordial dwarfism (MPD). However, while the study is compelling and well-constructed, additional experiments and discussions are needed to strengthen the conclusions.

Major comments

1. Endogenous MCM5 UFMylation.

While the authors convincingly demonstrate MCM5 UFMylation using overexpression systems (e.g., FLAG-MCM5 and HA-UFM1), the manuscript does not provide evidence that endogenous MCM5 is UFMylated under physiological conditions. This is a critical gap, as it raises concerns about the relevance of the observed UFMylation in the context of endogenous cellular processes.

• Suggestions:

1. Immunoprecipitation of Endogenous MCM5:

Perform immunoprecipitation (IP) of endogenous MCM5 from untreated cells and evaluate its UFM1 modification status using specific anti-UFM1 antibodies. Include replication stress conditions (e.g., hydroxyurea or aphidicolin treatment) to assess whether UFMylation is regulated dynamically.

2. Chromatin-bound MCM5:

Analyze UFM1 modification on chromatin-bound MCM5 during DNA replication using chromatin fractionation followed by IP and Western blotting. This would provide evidence of UFMylation occurring in the context of the replisome.

3. Loss-of-function Analysis:

Assess whether MCM5 UFMylation is abolished in cells lacking key UFM1 pathway components (e.g., UFL1, UFC1, or UFBP1 knockdown).

Without confirming that MCM5 is endogenously UFMylated, the physiological relevance of the findings remains questionable and needs to be addressed.

2. UFSP2 and Reversible UFMylation

The manuscript demonstrates that UFSP2, a UFM1-specific peptidase, can remove UFM1 modifications from MCM5, highlighting the reversible nature of UFMylation. However, the functional consequences of UFSP2 deficiency on DNA replication and repair have not been fully explored.

• Suggestions:

1. DNA Repair Defects in UFSP2 Deficiency:

Investigate whether UFSP2 deficiency leads to defects in DNA repair. For example, assess γ H2AX foci formation or chromosomal aberrations in cells lacking UFSP2 under replication stress conditions (e.g., aphidicolin or hydroxyurea treatment).

2. Functional Impact on DNA Replication:

Perform DNA fiber assays to evaluate whether UFSP2 deficiency affects replication fork progression or stability. This could reveal whether the dynamic regulation of UFMylation by UFSP2 is critical for replisome function.

3. UFSP2 Localization and Recruitment:

Examine the localization of UFSP2 during the cell cycle, particularly at replication forks or chromatin, to determine whether it functions locally to reverse UFMylation.

3. UFBP1 and CDK5RAP3

While the manuscript convincingly demonstrates the role of UFMylation on MCM5, it does not address the involvement of UFBP1 and CDK5RAP3, key components of the UFM1 E3 ligase complex. Both are known to interact with UFM1-modified substrates in other systems (e.g., ER-RQC) and may play a role in regulating MCM5 UFMylation or its function in DNA replication.

• Suggestions:

1. Perform knockdown or knockout of UFBP1 and CDK5RAP3 to evaluate their roles in MCM5 UFMylation and DNA replication.
2. Include a discussion on their potential involvement in recognizing UFMylated MCM5 or facilitating its function in CMG helicase stability.

4. Subcellular Localization Analysis

While the study provides functional data on MCM5 UFMylation, it lacks information on the spatiotemporal dynamics of UFM1-related factors (UBA5, UFC1, UFL1, UFBP1, CDK5RAP3). Understanding their localization during the cell cycle is crucial to further clarifying their roles in DNA replication.

• Suggestions:

1. Immunofluorescence (IF) Localization Studies:

Perform IF analysis of UFM1-related factors to determine their localization at different stages of the cell cycle, particularly their presence at replication origins or active replisomes. Assess whether localization is altered in conditions that inhibit UFMylation (e.g., UFL1 knockdown or DKM 2-93 treatment).

2. Dynamic Localization and Recruitment:

Investigate whether UFM1 E3 ligase components are dynamically recruited to chromatin or replisomes during DNA replication. Use synchronization techniques (e.g., nocodazole or thymidine block) to capture temporal changes in localization.

3. Functional Correlation:

Combine localization data with functional assays (e.g., EdU incorporation or DNA fiber assays) to correlate the localization of UFM1-related factors with DNA replication efficiency.

5. Direct Link to MPD Pathogenesis

The manuscript links UFMylation pathway mutations to MPD but lacks direct evidence showing that MCM5 UFMylation defects contribute to MPD pathogenesis.

Minor Comments

1. Figures:

- Include molecular weight markers in all Western blot images (e.g., Figure 4C, 4D, S5E).
- Ensure all figure legends provide sufficient detail, including experimental conditions (e.g., for Figure 6C).

Referee #2:

In this study, S. Takeda, X. Xu and co-workers study the effect of UFMylation on DNA replication. This post-translational modification of target proteins by an ubiquitin-like protein (UFM) plays a role in several cellular processes, including the DNA damage response. Mutations in the E1-E2-E3 proteins responsible for UFMylation (UBA5, UFC1 and UFL1) have been linked to microcephalic primordial dwarfism disorders such as Meier-Gorlin syndrome (MGS). Because MGS can also be caused by mutations in DNA replication genes (e.g. MCMs, CDC45 and GINS), the authors hypothesize that UFMylation is mechanistically linked to DNA replication.

Using EdU incorporation assays and stretched DNA fiber analysis, they find that UFMylation is required to maintain normal levels of origin firing and fork progression rate (Figures 1 and S1). Via iPOND and chromatin immunoprecipitation, UFL1 is found at forks and origins, influencing their activation time (Figures 2 and S2). Immunoprecipitation, PLA and other biochemical approaches reveal that UFL1 interacts with CMG helicase components (Figure 3 and S3). MCM5 is found as a direct target of UFMylation in the context of replication forks and origins (Figures 4, S4 and 5). K583 residue in MCM5 is identified as an UFM-binding site (Figure S5). Mutation of this residue affects CMG stability, origin activity and fork progression (Figures 6 and S6). Finally, interference with MCM5 UFMylation results in signs of chromosomal instability (Figures 7 and S7).

The regulation of DNA replication in both primary and cancer cells is a very active field of research. This manuscript is clearly written, and the experiments are neatly designed and performed. Furthermore, the study introduces UFMylation as a novel element in the already complex pathways that regulate replication origin licensing and activation. Based on the combination of novelty and experimental quality, I think the article merits publication after the authors have addressed a few points:

1. In Figure 1C, the concentration of UBA5 inhibitor DKM 2-93 is not indicated.
2. The analysis of inter-origin distance (Figure 4b) is not sufficiently described in Materials and Methods. What technique was used in Figures 1D and Figure 6A? Stretched DNA fibers or actual DNA combing? DNA combing is not mentioned, but in the text (p. 7), the "DNA fiber assay" is mentioned for the first time after the IOD experiment has been presented. I am asking because several laboratories have experienced that the use of stretched DNA fibers to monitor IOD is challenging, as the average fiber length is shorter than with DNA combing. Important aspects to be indicated are the total number of structures counted in each case, and whether the authors have used a DNA antibody or other method to monitor fiber integrity and make sure that they are counting adjacent origins in the same DNA molecule.
3. In Figure 1G (phenotypic rescue of UFM1 depletion with ectopic expression of WT but not mutant versions of UFM1). Immunoblots with the expression levels of the WT and the different mutants should be shown as a control of the experiment.
4. The results shown in Figure 2A and Figure S2A-D are very interesting but also seem slightly at odds with each other. First, the data in Figure S2 clearly shows that UFL1 is present on chromatin at all stages of the cell cycle, similar to a histone or a stable component of the ORC complex. On the other hand, the iPOND experiment in Figure 2A suggests otherwise: UFL1 is clearly recovered from the EdU pulse sample (actual forks) but not from the EdU+Thy chase sample (mature chromatin). What is the interpretation of the authors? It would be informative if the levels of UFL1 across the cell cycle (Figure S2) were not shown only in the chromatin fraction but also in the soluble fraction and in whole cell extracts. This would offer a more complete picture of the (possibly subtle) cell cycle fluctuations in UFL1 binding to DNA.
5. Related to previous point; the biochemical fractionation method is described in Materials and Methods as "previously described (67)". It should be noted that ref 67 uses exactly the same protocol published before by the Stillman laboratory (PMID: 11046155; not cited in either study). This is quite obvious in Figure S4A that uses the fraction nomenclature (S2, S3, P3) from the original article.
6. A central message of this study (as shown in Figures 4 and 5) is the presumed UFMylation of MCM5 at residue Lys583. I agree that several lines of indirect evidence point to it, e.g. the phenotypes described when K583 is mutated. Could the authors prove directly that the modifications detected in MCM5 protein in HEK293T cells (e.g. Figure 4A, 4C or 4D) correspond to UFMylation? This could be tried by IP (or pulldown) followed by mass spectrometry.

OTHER MINOR POINTS

7. Revise sentence in p. 13, l. 3 ("UBA5 restrains exposure restrains replication...")

8. The authors should tone down or qualify the claim that "UFL1 is present at replication origins", as this has been only shown by ChIP enrichment in two origins, out of the many thousands of origins described as ORC-binding sites (or SNS-Seq) in mammalian cells.
9. p. 21, first line of last paragraph. The sentence "We next tested whether MCM5 de-UFMylation was constitutive" is confusing.
10. Figure S5: the legend title does not correspond to the experiments in the Figure.
11. In p. 34, the authors seem to state that they will use HeLa cancer cells to address the final set of experiments related to genome instability, but HeLa cells have been used throughout the study.

Referee #3:

- general summary and opinion about the principal significance of the study, its questions and findings

In this study, Li et al demonstrated that ubiquitin-like modification (UFMylation) toward MCM5, a subunit of MCM2-7 complex, promotes CMG (CDC45-MCM-GINS) helicase assembly at replication forks and ensures a normal replication fork progression rate. Although the detailed mechanism is still unclear, this is probably the first report showing the involvement of UFMylation in the regulation of DNA replication. I think this work is novel enough to be published in the EMBO Journal. Below, I suggest some feasible experiments to support the conclusions and deepen the understanding of the role of UFL1 before publication.

- specific major concerns essential to be addressed to support the conclusions

1. In Figure S5, D1M, D3M, D5M and D7M mutants were also poorly UFMylated, suggesting that there may be additional UFMylation sites other than K583. Since the phenotypes of K583R alone and K583R+DKM2-93 were similar (Fig. 6), it is unlikely that UFMylation on other sites may support DNA replication activity. Nevertheless, it is desirable to clarify to what extent MCM5 UFMylation on chromatin (or in the replisome) was suppressed when K583R was expressed in the MCM5-depleted cells with siRNA as in the experiments with Fig. S4.
2. In Fig. 2, it was shown using iPOND and ChIP that UFL1 is present at origins and ongoing forks. This means that UFL1 is recruited to origins at some point during the cell cycle and travels with replication fork after origin firing. On the whole chromatin level, UFL1 is present throughout the cell cycle (Fig. S2). It will be important to clarify when and how UFL1 is recruited to origins. Is it possible to show the timing of UFL1 recruitment onto origins using ChIP assay with G1, G1/S, and G2/M cells (e.g., Nocodazole arrest/release cells)?
3. In Fig. S4B & S4C, UFL1 associates with and UFMyates MCM5 at 8hr after Nocodazole release. To better understand the mechanism of DNA replication initiation in higher eukaryotes, it is valuable to show whether those events (UFL1-MCM5 interaction and MCM5 UFMylation) are dependent on S-CDK and/or DDK (CDC7) kinase activities.

- minor concerns that should be addressed

1. Overall, the molecular weight marker should be added to the western blotting data. In addition, there are many descriptions about quantity on the blot (such as 50% decrease, 2-fold reduction etc.) Are they based on proper quantification?
2. In Fig. 3F and S3E, it is described "--- DKM 2-93 or siUFL delayed the loading of GINS3 by 2h ---. If the GINS loading reached normal level at 10h (not examined), it was "delay". But in the present case, it seems that the initiation efficiency was reduced while the initiation timing was not drastically delayed. Besides, the quality of anti-CDC45 antibody may not be good enough, since the protein was detected on chromatin at 0 h after Noc release and the bands appeared as doublet.
3. In the subtitle on the page 8, there is a misspelling: MPD eausative, MPD causative.
4. On the page 13 (line 3), there is an overlapping expression "--- restrains exposure restrains ---."

- any additional non-essential suggestions for improving the study (which will be at the author's/editor's discretion)

1. In the part of Abstract, there is an expression that "---, delaying origin firing by two hours and slowing replication fork progression by 70%." These specific values (two hours and 70%) may be deleted because they are not universal and variable depending on the cell type or conditions.
2. In the part of Introduction, it was described that "--- that mutations in DONSON associated with MPD typically affect its dimerization." But it is not known whether dimerization mutation is typical or not (MPD mutants other than Trp228Leu and Met446Thr have not been tested).
3. Some explanation about MCM10 may be added in Fig. 5B.

Point-by-point responses:

Referee #1:

General Comments

This manuscript explores the role of UFMylation, specifically MCM5 UFMylation at Lys583, in DNA replication, with a focus on its impact on replication origin firing and fork progression. The study provides important insights into the mechanistic link between UFMylation and the CMG helicase complex, highlighting its relevance in diseases such as microcephalic primordial dwarfism (MPD). However, while the study is compelling and well-constructed, additional experiments and discussions are needed to strengthen the conclusions.

Response: Thank you for the interest in our study, and the helpful comments raised that have enabled us to elevate our manuscript further.

Major comments

Q1. Endogenous MCM5 UFMylation.

While the authors convincingly demonstrate MCM5 UFMylation using overexpression systems (e.g., FLAG-MCM5 and HA-UFM1), the manuscript does not provide evidence that endogenous MCM5 is UFMylated under physiological conditions. This is a critical gap, as it raises concerns about the relevance of the observed UFMylation in the context of endogenous cellular processes.

Response: Thank you for highlighting this important point. We would like to note that the previous version of our manuscript already demonstrated endogenous MCM5 UFMylation under physiological conditions, using a specific anti-UFM1 antibody (Original Figures S4D–G; see below). These data show that MCM5 UFMylation occurs naturally within the replisome, independent of any overexpression systems.

Following the second suggestion, we replaced the original version of Figure 4C (western blot analysis of cells overexpressing a UFM1 transgene) with a revised version of Figure 4C (western blot analysis of cells expressing endogenous UFM1 alone); please see our response to the second suggestion below for more detail.

We also agree with the editor's comment that 'it will be important to see whether better emphasizing them would help to convince this referee'. In light of this, we changed the subtitle of the result (pages 14) and the Figure S4 title from 'MCM5 UFMylation occurs at the ongoing replication fork' to 'The UFMylation of MCM5 occurs at the ongoing replication fork'

Original Figure S4D-G

- Suggestions:

1. Immunoprecipitation of Endogenous MCM5:

Perform immunoprecipitation (IP) of endogenous MCM5 from untreated cells and evaluate its UFM1 modification status using specific anti-UFM1 antibodies. Include replication stress conditions (e.g., hydroxyurea or aphidicolin treatment) to assess whether UFMylation is regulated dynamically.

Response: Thank you for this suggestion. As requested, we performed immunoprecipitation of endogenous MCM5 from untreated 293T cells and assessed its UFM1 modification using an anti-UFM1 antibody (Graphic 1; see below). We also examined MCM5 UFMylation under replication stress conditions (2 mM HU treatment for 1 hour) and observed that this treatment had no significant effect on UFMylation levels. We have not included the data below because investigating MCM5 UFMylation under stress conditions is beyond the scope of our work.

Graphic 1

2. Chromatin-bound MCM5:

Analyze UFM1 modification on chromatin-bound MCM5 during DNA replication using chromatin fractionation followed by IP and Western blotting. This would provide evidence of UFMylation occurring in the context of the replisome.

Response: We appreciate this suggestion and have incorporated the requested data (see the data below) in Figure 4C. The original version of Figure 4C showed the UFMylation of chromatin-bound endogenous MCM5 in cells overexpressing UFM1 and UFMylation enzymes, whereas revised Figure 4C now shows the UFMylation of chromatin-bound endogenous MCM5 in cells not expressing UFMylation factors. In summary, the data presented in revised Figure 4C, in addition to CMG-IP, and iPOND findings (original Figure S4D-G) consistently support that endogenous MCM5 UFMylation occurs at active replication forks.

Revised Figure 4C

3. Loss-of-function Analysis:

Assess whether MCM5 UFMylation is abolished in cells lacking key UFM1 pathway components (e.g., UFL1, UFC1, or UFBP1 knockdown).

Response: Thank you for the suggestion. In response, we performed an experiment involving UFL1 knockdown in HEK293T cells (Graphic 2; see below). The results show that UFL1 depletion significantly reduces endogenous MCM5 UFMylation, further supporting the role of UFL1 in mediating this modification.

Graphic 2

Without confirming that MCM5 is endogenously UFMylated, the physiological relevance of the findings remains questionable and needs to be addressed.

Response: We agree with this point. As shown in Figures S4D–G and noted above, we demonstrated that MCM5 undergoes endogenous UFMylation under physiological conditions. In addition, as requested, we have incorporated new data into revised Figure 4C showing the UFMylation of chromatin-bound endogenous MCM5 using a specific anti-UFM1 antibody. We believe these results directly support the physiological relevance of our findings.

Q2. UFSP2 and Reversible UFMylation

The manuscript demonstrates that UFSP2, a UFM1-specific peptidase, can remove UFM1 modifications from MCM5, highlighting the reversible nature of UFMylation. However, the functional consequences of UFSP2 deficiency on DNA replication and repair have not been fully explored.

Response: Thank you for this thoughtful suggestion; however, we believe that analyzing the functional consequences of UFSP2 deficiency is not an appropriate approach for investigating 'the reversible nature of UFMylation' because UFSP2 not only removes

UFMI from UFMylated proteins, as a UFM1-specific peptidase, but is also required for UFM1 maturation. Furthermore, we believe that investigating the impact of UFSP2 deficiency on DNA repair is beyond the scope of this study. In order to investigate the role of UFSP2 in DNA replication, we overexpressed UFSP2 in HeLa cells and found the replication speed was reduced in a catalytic-dependent way (Graphic 3).

• Suggestions:

1. DNA Repair Defects in UFSP2 Deficiency:

Investigate whether UFSP2 deficiency leads to defects in DNA repair. For example, assess γ H2AX foci formation or chromosomal aberrations in cells lacking UFSP2 under replication stress conditions (e.g., aphidicolin or hydroxyurea treatment).

Response: Please refer to our response to the previous query.

2. Functional Impact on DNA Replication:

Perform DNA fiber assays to evaluate whether UFSP2 deficiency affects replication fork progression or stability. This could reveal whether the dynamic regulation of UFMylation by UFSP2 is critical for replisome function.

Response: Please refer to our response to the previous query. Nonetheless, to address the dynamic regulation of UFMylation by UFSP2, we assessed the impact of UFSP2 overexpression on DNA replication in HeLa cells and measured replication speed by DNA fiber assay (Graphic 3; see below). As with UFL1 depletion, overexpression of wild-type UFSP2, but not a catalytic-dead mutant, reduced replication speed and markedly decreased MCM5 UFMylation (Figure 4F in the revised manuscript). These findings support the concept that the dynamic regulation of UFMylation by UFSP2 plays a critical role in DNA replication.

Graphic 3

3. UFSP2 Localization and Recruitment:

Examine the localization of UFSP2 during the cell cycle, particularly at replication forks or chromatin, to determine whether it functions locally to reverse UFMylation.

Response: Thank you for this suggestion. We performed a chromatin fractionation (*Revised Figures S2A and S2B*; see below) and an immunofluorescence assay to examine UFSP2 localization throughout the cell cycle (*Graphic 4*). We observed that chromatin-bound UFSP2 decreases from the M phase to early G1 (0 to 4 hours), increases slightly from early to late G1, and then decreases modestly during the S phase (*Revised Figures S2A and S2B*).

Revised Figure S2A and S2B

To further assess its localization, we co-stained UFSP2 with EdU and Cyclin A to distinguish between cells in the G1, S, and G2 phases (*Graphic 4*). We found that UFSP2's nuclear localization remains largely unchanged from G1 through G2, consistent with our chromatin fractionation results. By contrast, UFSP2 is specifically enriched at the centrosome and spindle during metaphase, suggesting its potential role in mitosis.

Q3. UFBP1 and CDK5RAP3

While the manuscript convincingly demonstrates the role of UFMylation on MCM5, it does not address the involvement of UFBP1 and CDK5RAP3, key components of the UFM1 E3 ligase complex. Both are known to interact with UFM1-modified substrates in other systems (e.g., ER-RQC) and may play a role in regulating MCM5 UFMylation or its function in DNA replication.

Response: Thank you for your constructive feedback and your positive conclusion that 'the manuscript convincingly demonstrates the role of UFMylation on MCM5'. As the reviewer appreciated, our study uncovered the role of UFMylation in physiological DNA replication. As discussed and agreed upon with the editor, the regulation of the UFM1 E3 ligase complex is an important issue of future studies but is beyond the scope of this study.

- Suggestions:

1. Perform knockdown or knockout of UFBP1 and CDK5RAP3 to evaluate their roles in MCM5 UFMylation and DNA replication.

Response: Thank you for the suggestion. As noted above, we believe that the regulation of the UFM1 E3 ligase complex is beyond the scope of this manuscript but will be an interested area for future research.

2. Include a discussion on their potential involvement in recognizing UFMylated MCM5 or facilitating its function in CMG helicase stability.

Plan: Thank you for raising this important point. At present, we are unable to provide a structural discussion of how the E3 complex recognizes MCM5 for UFMylation due to the lack of data involving UFBP1 and CDK5RAP3.

Q4. Subcellular Localization Analysis

While the study provides functional data on MCM5 UFMylation, it lacks information on the spatiotemporal dynamics of UFM1-related factors (UBA5, UFC1, UFL1, UFBP1, CDK5RAP3). Understanding their localization during the cell cycle is crucial to further clarifying their roles in DNA replication.

- Suggestions:

1. Immunofluorescence (IF) Localization Studies:

Perform IF analysis of UFM1-related factors to determine their localization at different stages of the cell cycle, particularly their presence at replication origins or active

replisomes. Assess whether localization is altered in conditions that inhibit UFMylation (e.g., UFL1 knockdown or DKM 2-93 treatment).

Response: Thank you for your suggestions. We performed immunostaining to assess the localization of UFM1, UBA5, and UFL1 in HeLa cells treated with either DMSO or DKM 2-93 throughout the cell cycle (Graphic 5; see below). We observed that the nuclear signals for UFM1 and UFL1 gradually increase from G1 to S phase, peaking in G2, whereas UBA5 localization remains unchanged from G1 through G2. DKM 2-93 treatment did not significantly affect UFM1 localization, likely because the detected UFM1 signal includes both conjugated and free forms. Similarly, UFL1 localization was largely unaffected from G1 to G2 phase, although we observed a marked reduction in its signal during mitosis; this is an unexpected finding that warrants further investigation in future studies. UBA5 localization was also unaffected by DKM 2-93, consistent with the compound's role as an activity inhibitor rather than a localization disruptor.

Graphic 5

To further investigate the chromatin association of UFMylation components, we conducted immunofluorescence analyses using the CSK pre-extraction method and included DKM 2-93 treatment (**Graphic 6**; see below). Co-staining with MCM5 allowed us to assess colocalization at chromatin. The data showed that UFM1, UFL1, CDK5RAP3, and UFBP1 are all present on chromatin and co-localize with chromatin-bound MCM5 during S phase, supporting a role in DNA replication. Notably, DKM 2-93 treatment for 1 hour reduced the chromatin-associated UFM1 signal but had no significant effect on the chromatin localization of UFL1, CDK5RAP3, or UFBP1. This finding suggests that the chromatin recruitment of the E3 ligase complex is independent of active UFMylation.

Graphic 6

2. Dynamic Localization and Recruitment:

Investigate whether UFM1 E3 ligase components are dynamically recruited to

chromatin or replisomes during DNA replication. Use synchronization techniques (e.g., nocodazole or thymidine block) to capture temporal changes in localization.

Response: Thank you for your suggestion. We performed a chromatin fractionation assay to assess the localization of the UFL1–CDK5RAP3–UFBP1 complex in both soluble and chromatin fractions following nocodazole release (**Revised Figures S2A and S2B**; see below). The chromatin-bound UFL1 signal decreased between 0 and 4 hours (corresponding to the transition from M phase to early G1 phase) and then increased from 4 to 8 hours, coinciding with the loading of GINS3 onto chromatin. By contrast, UFBP1 and CDK5RAP3 remained stably associated with chromatin throughout the M to S phase transition, showing no marked temporal fluctuations.

Revised Figure S2A and S2B

3. Functional Correlation:

Combine localization data with functional assays (e.g., EdU incorporation or DNA fiber assays) to correlate the localization of UFM1-related factors with DNA replication efficiency.

Response: Based on the revised data above, it is challenging to directly correlate the subcellular localization patterns of the E3 complex (UFL1, UFBP1, and CDK5RAP3) with functional outcomes.

Q5. Direct Link to MPD Pathogenesis

The manuscript links UFMylation pathway mutations to MPD but lacks direct evidence showing that MCM5 UFMylation defects contribute to MPD pathogenesis.

Response: Thank you for this comment. In our opinion, providing direct evidence linking MCM5 UFMylation defects to MPD pathogenesis would require animal models, which often fail to fully recapitulate the neurological features observed in patients. We thus believe that the best way to address this issue is to analyze the correlation between mutations in DNA replication machinery and MPD pathogenesis, as exemplified in previous clinical studies [PMID: 21358632, 21358633, 26637980, 27374770, 21358631, 33654309]. By showing the role of UFMylation in DNA replication, our studies support the idea that hypomorphic mutations in UFMylation enzymes likely cause MPD by impairing DNA replication.

Minor Comments

1. Figures:

- Include molecular weight markers in all Western blot images (e.g., Figure 4C, 4D, S5E).

Response: Thank you for your suggestion; we have revised the figures as required.

- Ensure all figure legends provide sufficient detail, including experimental conditions (e.g., for Figure 6C).

Response: Thank you for your suggestion. We have revised the legends as required, and in line with the journal's formatting guidelines.

Referee #2:

In this study, S. Takeda, X. Xu and co-workers study the effect of UFMylation on DNA replication. This post-translational modification of target proteins by an ubiquitin-like protein (UFM) plays a role in several cellular processes, including the DNA damage response. Mutations in the E1-E2-E3 proteins responsible for UFMylation (UBA5, UFC1 and UFL1) have been linked to microcephalic primordial dwarfism disorders such as Meier-Gorlin syndrome (MGS). Because MGS can also be caused by mutations in DNA replication genes (e.g. MCMs, CDC45 and GINS), the authors hypothesize that UFMylation is mechanistically linked to DNA replication.

Using EdU incorporation assays and stretched DNA fiber analysis, they find that UFMylation is required to maintain normal levels of origin firing and fork progression rate (Figures 1 and S1). Via iPOND and chromatin immunoprecipitation, UFL1 is found at forks and origins, influencing their activation time (Figures 2 and S2). Immunoprecipitation, PLA and other biochemical approaches reveal that UFL1 interacts with CMG helicase components (Figure 3 and S3). MCM5 is found as a direct target of UFMylation in the context of replication forks and origins (Figures 4, S4 and 5). K583 residue in MCM5 is identified as an UFM-binding site (Figure S5). Mutation of this residue affects CMG stability, origin activity and fork progression (Figures 6 and S6). Finally, interference with MCM5 UFMylation results in signs of chromosomal instability (Figures 7 and S7).

The regulation of DNA replication in both primary and cancer cells is a very active field of research. This manuscript is clearly written, and the experiments are neatly designed and performed. Furthermore, the study introduces UFMylation as a novel element in the already complex pathways that regulate replication origin licensing and activation. Based on the combination of novelty and experimental quality, I think the article merits publication after the authors have addressed a few points:

Response: Thank you for the comments.

1. In Figure 1C, the concentration of UBA5 inhibitor DKM 2-93 is not indicated.

Response: We apologize for this oversight and have added the concentration (100 μ M) to the corresponding figure legend.

2. The analysis of inter-origin distance (Figure 4b) is not sufficiently described in Materials and Methods. What technique was used in Figures 1D and Figure 6A? Stretched DNA fibers or actual DNA combing? DNA combing is not mentioned, but in

the text (p. 7), the “DNA fiber assay” is mentioned for the first time after the IOD experiment has been presented. I am asking because several laboratories have experienced that the use of stretched DNA fibers to monitor IOD is challenging, as the average fiber length is shorter than with DNA combing. Important aspects to be indicated are the total number of structures counted in each case, and whether the authors have used a DNA antibody or other method to monitor fiber integrity and make sure that they are counting adjacent origins in the same DNA molecule.

Response: We apologize for our insufficient description in the Materials and Methods. We used stretched DNA fibers to generate the data shown in Figures 1D and 6A. We have added the detailed method for deriving the “Inter-origin distance (IOD)” to the Materials and Methods, as follows:

“Replication origins are identified by red–green–red fiber patterns, where a green segment (CldU-labeled) is flanked by red segments (IdU-labeled) on both sides. Occasionally, the green segment may appear split by an unstained region; this reflects bidirectional fork movement away from the origin during CldU incorporation. Only fibers lying exactly on a horizontal line are considered to be from the same DNA sample and are counted. The midpoint between two adjacent green segments is taken as the origin location. The IOD is then calculated by measuring the linear distance between these origins along the same DNA fiber, with values expressed in micrometers to reflect the physical spacing between replication origins during DNA synthesis.”

*We agree that ‘the use of stretched DNA fibers to monitor IOD is challenging, as the average fiber length is shorter than with DNA combing’. Nonetheless, it is possible to compare the IOD between wild-type and MCM5-UFMylation-deficient cells in **revised Figures 1D and Figure 6A** (see below). Doing so allowed us to conclude an ~40% increase in IOD in the MCM5-UFMylation-deficient cells. We agreed with the comment that the total number of structures in each case must be counted. We have now added*

the total number to revised **Figures 1D** and **6B** (see below), and changed revised **Figure 6B** from a histogram to a violin plot.

Revised Figure 6A and 6B

We also agree that we have to make sure to count adjacent origins in the same DNA molecule. Following the reviewer's suggestion, we repeated the experiment in Figure 1D, using a DNA antibody in addition to anti-CldU and anti-IdU antibodies (**Graphic 7A**; see below). In doing so, we confirmed that two neighboring replication origins to measure IOD are localized on the same DNA (**Graphic 7A**; see below) and reproduced the same data (**Graphic 7B**; see below) as provided in original Figure 1D. We have decided not to provide Response Figure 7 in our revised manuscript because the quality of CldU-labeling dropped during the multi-step staining of the stretched DNA fibers. Please note that the number of CldU/IdU-labeled DNA was low (**Graphic 8**; see below), and it is unlikely that we counted adjacent origins in different DNA molecules even without DNA staining.

Q3. In Figure 1G (phenotypic rescue of UFM1 depletion with ectopic expression or WT but not mutant versions of UFM1). Immunoblots with the expression levels of the WT and the different mutants should be shown as a control of the experiment.

Response: We apologize for this oversight. The relevant western blot data have been added to the revised manuscript (Revised Figures S1N-P; see below).

Revised Figures S1N-P

Q4. The results shown in Figure 2A and Figure S2A-D are very interesting but also seem slightly at odds with each other. First, the data in Figure S2 clearly shows that UFL1 is present on chromatin at all stages of the cell cycle, similar to a histone or a stable component of the ORC complex. On the other hand, the iPOND experiment in Figure 2A suggests otherwise: UFL1 is clearly recovered from the EdU pulse sample (actual forks) but not from the EdU+Thy chase sample (mature chromatin). What is the interpretation of the authors? It would be informative if the levels of UFL1 across the cell cycle (Figure S2) were not shown only in the chromatin fraction but also in the soluble fraction and in whole cell extracts. This would offer a more complete picture of the (possibly subtle) cell cycle fluctuations in UFL1 binding to DNA.

Response: We appreciate this insightful observation. It is not unexpected that the iPOND data (Original Figure 2A; see below) and the chromatin fractionation data (Original Figure S2A; see below) appear to differ. The data from our iPOND

experiment suggest that UFL1 is part of the active replication machinery, and the western blot data of the chromatin fraction suggests a substantial fraction of UFL1 is constantly associated with chromatin proteins irrespective of cell cycle phases.

Original Figure 2A and S2A

To clarify these observations, and following the reviewer's suggestion, we reanalyzed UFL1 protein levels, examining its presence in the whole cell extract and the soluble and chromatin fractions across the cell cycle. We found that UFL1 levels remain stable in the whole cell extract and the soluble fraction, while its chromatin association changes dynamically during the cell cycle (**Revised Figure S2A-S2D**; see below). Additionally, we observed a slight increase in UFL1 nuclear foci during the S phase, as shown in the revised data (**Graphic 5**; see above) in our response to Reviewer #1, Q4-1. Furthermore, new data from experiments prompted by Reviewer #3 (Q2) demonstrate that UFL1 specifically binds to the LMNB2 origin during the G1/S transition and S phase (**Revised Figure 3F and 3G**; see below). Taken together, these results offer a more comprehensive picture of UFL1's dynamic association with chromatin and its cell cycle-dependent recruitment to replication origins.

Revised Figure S2A-S2D

Revised 3F and 3G

Q5. Related to previous point; the biochemical fractionation method is described in Materials and Methods as “previously described (67)”. It should be noted that ref 67 uses exactly the same protocol published before by the Stillman laboratory (PMID: 11046155; not cited in either study). This is quite obvious in Figure S4A that uses the fraction nomenclature (S2, S3, P3) from the original article.

Response: Thank you for bringing this to our attention. In this study, we used two distinct methods for component separation. The approach used in Figure S4 corresponds to the protocol described in the reference you cited: *Mol Cell Biol* 2000 (PMID: 11046155). For all other chromatin fractionation experiments in the manuscript, we followed the protocol published in *Molecular Cell* 2004 (PMID: 15149598). Both references are now appropriately cited in the revised manuscript to clearly indicate the methodologies applied.

Q6. A central message of this study (as shown in Figures 4 and 5) is the presumed UFMylation of MCM5 at residue Lys583. I agree that several lines of indirect evidence point to it, e.g. the phenotypes described when K583 is mutated. Could the authors prove directly that the modifications detected in MCM5 protein in HEK293T cells (e.g. Figure 4A, 4C or 4D) correspond to UFMylation? This could be tried by IP (or pulldown) followed by mass spectrometry.

Response: We appreciate this suggestion. We have data showing that the modification signal is completely abolished when HA-UFM1 is replaced with an HA-vector, supporting the conclusion that the observed modification represents UFMylation (**Graphic 9A**; see below). Additionally, we enriched UFM1 by denaturing immunoprecipitation and detected MCM5 in the resulting mass spectrometry data (data not shown).

We do acknowledge, however, that identifying UFMylation sites by mass spectrometry is technically challenging. This is largely due to the unique C-terminal CG residues of mature UFM1, which differ from the GG residues found in ubiquitin and other ubiquitin-like proteins, thereby limiting detection efficiency. Furthermore, the high abundance of unmodified MCM5 in both the chromatin and cytoplasmic fractions makes it even more difficult to isolate and characterize the UFMylated form. We have repeated the IP–mass spectrometry assay to try to directly confirm the MCM5 UFMylation site (**Graphic 9B**; see below), but given the inherent technical limitations, we have not been successful thus far.

Graphic 9

A

B

OTHER MINOR POINTS

7. Revise sentence in p. 13, l. 3 (“UBA5 restrains exposure restrains replication...”)

Response: Thank you for spotting this typographical error. It has been corrected in the revised manuscript.

8. The authors should tone down or qualify the claim that “UFL1 is present at replication origins”, as this has been only shown by ChIP enrichment in two origins, out of the many thousands of origins described as ORC-binding sites (or SNS-Seq) in mammalian cells.

Response: Thank you for your suggestion. We agree and have revised the text to read: “We identified the enrichment of UFL1 at two identified replication origins”.

9. p. 21, first line of last paragraph. The sentence “We next tested whether MCM5 de-UFMylation was constitutive” is confusing.

Response: We apologize for the confusion. In the revised manuscript, we have clarified this statement as follows: “We next tested whether MCM5 UFMylation is reversible”.

10. Figure S5: the legend title does not correspond to the experiments in the Figure.

Response: We apologize for this error. The legend title has been revised to: "MCM5 is UFMylated on Lys583".

11. In p. 34, the authors seem to state that they will use HeLa cancer cells to address the final set of experiments related to genome instability, but HeLa cells have been used throughout the study.

Response: Thank you for spotting this inappropriate statement. We have removed it in the revised manuscript.

Referee #3:

- general summary and opinion about the principal significance of the study, its questions and findings

In this study, Li et al demonstrated that ubiquitin-like modification (UFMylation) toward MCM5, a subunit of MCM2-7 complex, promotes CMG (CDC45-MCM-GINS) helicase assembly at replication forks and ensures a normal replication fork progression rate. Although the detailed mechanism is still unclear, this is probably the first report showing the involvement of UFMylation in the regulation of DNA replication. I think this work is novel enough to be published in the EMBO Journal. Below, I suggest some feasible experiments to support the conclusions and deepen the understanding of the role of UFL1 before publication.

- specific major concerns essential to be addressed to support the conclusions

Q1. In Figure S5, D1M, D3M, D5M and D7M mutants were also poorly UFMylated, suggesting that there may be additional UFMylation sites other than K583. Since the phenotypes of K583R alone and K583R+DKM2-93 were similar (Fig. 6), it is unlikely that UFMylation on other sites may support DNA replication activity. Nevertheless, it is desirable to clarify to what extent MCM5 UFMylation on chromatin (or in the replisome) was suppressed when K583R was expressed in the MCM5-depleted cells with siRNA as in the experiments with Fig. S4.

Response: Thank you for this suggestion. To clarify the extent to which the K583R mutation suppresses MCM5 UFMylation, we measured the association of HA-UFM1 with FLAG-MCM5 (wild-type and K583R mutant) in endogenous-MCM5-depleted cells (Revised Figure S5D; see below). As discussed in the revised manuscript, we found that the K583R mutant exhibits a reduced UFMylation signal on chromatin compared to wild-type MCM5.

Revised Figure S5D

Q2. In Fig. 2, it was shown using iPOND and ChIP that UFL1 is present at origins and ongoing forks. This means that UFL1 is recruited to origins at some point during the cell cycle and travels with replication fork after origin firing. On the whole chromatin level, UFL1 is present throughout the cell cycle (Fig. S2). It will be important to clarify when and how UFL1 is recruited to origins. Is it possible to show the timing of UFL1 recruitment onto origins using ChIP assay with G1, G1/S, and G2/M cells (e.g., Nocodazole arrest/release cells)?

Response: To clarify the timing of UFL1 recruitment to replication origins, we performed a ChIP-qPCR assay to examine its dynamic binding across different cell cycle phases, including G1, G1/S, S, G2, and M. To enrich HeLa cells in M phase, we treated them with 333 nM nocodazole for 16 hours and collected mitotic cells by shake-off. For G1 phase cells, these mitotic cells were sub-cultured into fresh medium for 4 hours. The G1/S transition was achieved via double thymidine block, and G2 phase cells were obtained using a two-step treatment with thymidine followed by a CDK1 inhibitor. All samples were collected for subsequent ChIP analysis.

The data showed that UFL1 binds to the LMNB2 origin during the G1/S transition and S phase (Revised Figure 3F and 3G; see below). This pattern aligns with the

observed increase in the UFL1–MCM5 interaction on chromatin and elevated MCM5 UFMylation at 8 hours after nocodazole release, as shown in **Original Figures S4B and S4C** (see data below). The data indicated that UFL1 is recruited onto origin at the G1/S boundary.

Revised 3F and 3G

Original Figure S4B and S4C

Q3. In Fig. S4B & S4C, UFL1 associates with and UFMylates MCM5 at 8hr after Nocodazole release. To better understand the mechanism of DNA replication initiation in higher eukaryotes, it is valuable to show whether those events (UFL1-MCM5 interaction and MCM5 UFMylation) are dependent on S-CDK and/or DDK (CDC7) kinase activities.

Response: Thank you for proposing these interesting experiments. We treated HeLa cells with a CDC7 inhibitor and found that both the UFL1–MCM5 interaction (Graphic 10A and 10B; see below) and MCM5 UFMylation (Graphic 10C; see below) are unaffected by CDC7 inhibition. These results suggest that UFL1 binding to MCM5 and the subsequent UFMylation of MCM5 occur independently of CDC7 kinase activity, indicating that these events precede CDC7-dependent regulation of DNA replication initiation.

Graphic 10

- minor concerns that should be addressed

1. Overall, the molecular weight marker should be added to the western blotting data. In addition, there are many descriptions about quantity on the blot (such as 50% decrease, 2-fold reduction etc.) Are they based on proper quantification?

Response: We apologize for this oversight. A molecular weight marker has been added to each western blot in the revised manuscript. Regarding the quantifications, these were performed by measuring the gray values of the blots and normalizing them to the actin or histone H3 signal. To improve clarity and avoid overinterpretation, we have

revised the descriptions of changes (e.g. '50% decrease', '2-fold reduction') to 'significant increase' or 'significant decrease' to provide more accurate information.

2. In Fig. 3F and S3E, it is described “--- DKM 2-93 or siUFL delayed the loading of GINS3 by 2h ---. If the GINS loading reached normal level at 10h (not examined), it was “delay”. But in the present case, it seems that the initiation efficiency was reduced while the initiation timing was not drastically delayed. Besides, the quality of anti-CDC45 antibody may not be good enough, since the protein was detected on chromatin at 0 h after Noc release and the bands appeared as doublet.

Response: Thank you for raising this point. We have revised the text to read: “DKM 2-93 or siUFL reduced the loading of GINS3.” Additionally, we repeated the western blot assay using the same samples but an alternative CDC45 antibody (Cdc45 (D7G6) Rabbit mAb #11881) and generated a clearer CDC45 band signal (**Revised Figures 3H and S3E**; see below).

Revised Figure 3H and S3E

3. In the subtitle on the page 8, there is a misspelling: MPD eausative, MPD causative.

Response: Thank you for spotting this error; we have amended this spelling in our revised manuscript.

4. On the page 13 (line 3), there is an overlapping expression “--- restrains exposure restrains ---.”

Response: Thank you for spotting this error; we have amended this phrase in our revised manuscript.

- any additional non-essential suggestions for improving the study (which will be at the author's/editor's discretion)

1. In the part of Abstract, there is an expression that “---, delaying origin firing by two hours and slowing replication fork progression by 70%.” These specific values (two hours and 70%) may be deleted because they are not universal and variable depending on the cell type or conditions.

Response: Thank you for raising this point; we have amended the text accordingly in our revised abstract.

2. In the part of Introduction, it was described that “--- that mutations in DONSON associated with MPD typically affect its dimerization.” But it is not known whether dimerization mutation is typical or not (MPD mutants other than Trp228Leu and Met446Thr have not been tested).

Response: We agree with this point and have updated the revised manuscript accordingly to state: “--- that mutations in DONSON associated with MPD might affect its dimerization”.

3. Some explanation about MCM10 may be added in Fig. 5B.

Response: Thank you for raising this suggestion. We have added the following description to the main text of the revised manuscript: “The interaction between MCM5 and MCM10, another important factor in origin firing, was also reduced in siUFL1-depleted HeLa cells (Figure 5B)”.

Dr. Xingzhi Xu
Shenzhen University Medical School
Cell Biology
1066 Xueyuan Blvd, Rm. A6-901
Shenzhen University Lihu Campus
Shenzhen, Guangdong 518055
China

5th Jun 2025

Re: EMBOJ-2024-119820R
MCM5 UFMylation regulates replication origin firing and fork progression

Dear Drs. Xu and Takeda,

Thank you for submitting your revised manuscript to The EMBO Journal. The three original referees have now assessed it once more, and I am pleased to say that they found the study substantially improved and most of their initial criticisms addressed. Still, they retain a few specific concerns that would require textual/presentational modifications and in some cases (ref 1 point ii, ref 2 point 1) additional control experiments/analyses. I am therefore inviting you to a final round of minor revision, after which we should hopefully be able to proceed further with eventual acceptance of the study.

When preparing your final revised version, please also make sure to take care of the following editorial points:

- Please remove synopsis image and text from the main manuscript file, and upload both of them individually.
- Please rename Table EV1, which is included in the text, into Table 2 (also the reference to it); or move it out of the main text and upload it as a separate Table EV1 file in DOCX or XLSX format.

I am therefore returning the manuscript to you for a final round of revision, hoping you will be able to address the remaining referee and editorial points in a straightforward manner. Please do not hesitate to contact me should you have any questions in this regard!

With kind regards,

Hartmut

- 1) Every manuscript requires a Data Availability section (even if only stating that no deposited datasets are included). Primary datasets or computer code produced in the current study have to be deposited in appropriate public repositories prior to resubmission, and reviewer access details provided in case that public access is not yet allowed. Further information: embopress.org/page/journal/14602075/authorguide#dataavailability
- 2) Each figure legend must specify
 - size of the scale bars that are mandatory for all micrograph panels
 - the statistical test used to generate error bars and P-values
 - the type error bars (e.g., S.E.M., S.D.)
 - the number (n) and nature (biological or technical replicate) of independent experiments underlying each data point
 - Figures may not include error bars for experiments with $n < 3$; scatter plots showing individual data points should be used instead.
- 3) Revised manuscript text (including main tables, and figure legends for main and EV figures) has to be submitted as editable text file (e.g., .docx format). We encourage highlighting of changes (e.g., via text color) for the referees' reference.

9) To facilitate reproducibility and cross-laboratory adoption of methodologies, please structure the Materials & Methods section as outlined in our guide to authors, including a completed Reagents and Tools Table that can be downloaded from our author guidelines as well (<https://www.embopress.org/page/journal/14602075/authorguide#structuredmethods>).

10) Digital image enhancement is acceptable practice, as long as it accurately represents the original data and conforms to community standards. If a figure has been subjected to significant electronic manipulation, this must be clearly noted in the figure legend and/or the 'Materials and Methods' section. The editors reserve the right to request original versions of figures and the original images that were used to assemble the figure. Finally, we generally encourage uploading of numerical as well as gel/blot image source data; for details see: embopress.org/page/journal/14602075/authorguide#sourcedata

In the interest of ensuring the conceptual advance provided by the work, we recommend submitting a revision within 3 months (3rd Sep 2025). Please discuss the revision progress ahead of this time with the editor if you require more time to complete the revisions. Use the link below to submit your revision:

Link Not Available

Referee #1:

The revised manuscript responds thoroughly to nearly all of Referee #1's general and major comments, supplying new IP-WB, chromatin-fractionation and DNA-fiber data that convincingly document endogenous MCM5 UFMylation and its reversible control by UFL1/UFSP2. Nevertheless, two aspects still require clarification before the conclusions are fully robust. (i) UFSP2 function. The authors' rationale that UFSP2 knockout is uninterpretable because it lowers mature UFM1 is inconsistent with current evidence showing that pro-UFM1 maturation is largely mediated by UFSP1, whereas UFSP2 acts primarily as the major de-UFMyrase; mature UFM1 remains abundant in UFSP2-null cells, and only the UFSP1/UFSP2 double-KO abolishes maturation (Ishimura et al., FEBS Lett 2016 and subsequent studies). The response text on p. 5 of the point-by-point therefore needs updating, ideally citing double-KO data to justify the experimental design and removing the outdated claim that UFSP2 is required for maturation. (ii) UFBP1 and CDK5RAP3 selectivity. Because these adaptors confer substrate specificity to the UFL1 ligase during ER-RQC-linked UFMylation, it remains essential to demonstrate that MCM5 UFMylation (and associated fork phenotypes) are unaffected by UFBP1 or CDK5RAP3 depletion. Straightforward siRNA/CRISPR loss-of-function or rescue experiments-showing unchanged MCM5 UFMylation and fork speed in adaptor-deficient cells, or explicitly presenting negative results-would pre-empt concerns that the same adaptors are required outside the ER context.

Referee #2:

This is a revised version of an earlier manuscript reporting MCM5 UFMylation and its effects on replication origin activation and fork progression, in which Z. Li et al have addressed my comments and criticisms. While most of the minor comments have been properly solved, the new results presented in response to two of my major comments still require some clarification.

1. Analyses of inter-origin distance: the authors provide a better description of the methodology (stretched DNA fibers). From this updated description it is clear that no DNA antibody was used to control for fiber integrity. Instead, the authors relied on "fibers lying exactly on a horizontal line" to be considered part of the same DNA molecule. This approach is hardly rigorous for the reliable quantification of inter-origin distances. In the rebuttal letter, they do show an experiment that combined CldU, IdU and ssDNA stainings (the recommended approach) that reproduces the longer IODs upon siUFL1. However, the CldU staining in this case is spotty and hard to integrate with the IdU staining, making the identification of origin positions rather arbitrary. I realize that IOD experiments with stretched fibers are technically challenging (this was the point of my original criticism), but precisely for this reason the ssDNA antibody control is very important. One alternative way to back up the notion that fewer origins are activated upon siUFL1 without doing new experiments would be to quantify the relative number of red-green-red tracks relative to the total number of red-green forks tracks.

2. In Figure 1G, I had asked to add the immunoblots with expression levels of the WT and the different mutants of UFM1, UFC1 and UBA5. They are shown as new panels N-O-P in Figure S1. The results with UFM1 and UFC1 are clear, but two aspects of the results with UBA5 are confusing. First, the siRNA does not seem to downregulate endogenous UBA5 in most cases (assuming it is the lower band under 50 KD). Second, the levels of one of the mutants (Q302Ter*) are not shown, most likely because this is a truncation mutant with a different molecular weight - but in this case a larger section of the gel could be shown. These issues need to be discussed and clarified.

Referee #3:

I think the revised manuscript is satisfactory for publication just by correcting a few minor points as follows.

1. In Figure 6B, Inter-origin Distances (μM) must be Inter-origin Distances (μm).
2. In Figure 7A, the layout of the experimental conditions in horizontal axis was not fitted with the graph.
3. In Figure EV2, it might be better to add some explanation about UFBP1 and UFSP2 (and CDK5RAP3).
4. In line 13 of page 18, "This reduction in the proliferation rate likely ---" should be "This reduction in the progression rate likely ---".

Point-by-point responses:**Referee #1:**

The revised manuscript responds thoroughly to nearly all of Referee #1's general and major comments, supplying new IP-WB, chromatin-fractionation and DNA-fiber data that convincingly document endogenous MCM5 UFMylation and its reversible control by UFL1/UFSP2. Nevertheless, two aspects still require clarification before the conclusions are fully robust. (i) UFSP2 function. The authors' rationale that UFSP2 knockout is uninterpretable because it lowers mature UFM1 is inconsistent with current evidence showing that pro-UFM1 maturation is largely mediated by UFSP1, whereas UFSP2 acts primarily as the major de-UFMyrase; mature UFM1 remains abundant in UFSP2-null cells, and only the UFSP1/UFSP2 double - KO abolishes maturation (Ishimura et al., FEBS Lett 2016 and subsequent studies). The response text on p. 5 of the point-by-point therefore needs updating, ideally citing double-KO data to justify the experimental design and removing the outdated claim that UFSP2 is required for maturation. (ii) UFBP1 and CDK5RAP3 selectivity. Because these adaptors confer substrate specificity to the UFL1 ligase during ER-RQC-linked UFMylation, it remains essential to demonstrate that MCM5 UFMylation (and associated fork phenotypes) are unaffected by UFBP1 or CDK5RAP3 depletion. Straightforward siRNA/CRISPR loss-of-function or rescue experiments-showing unchanged MCM5 UFMylation and fork speed in adaptor-deficient cells, or explicitly presenting negative results-would pre-empt concerns that the same adaptors are required outside the ER context.

Response: Thank you for the helpful comments.

We agree the (i) statement. We removed 'the outdated claim that UFSP2 is required for maturation' and cited the UfSP1/UfSP2 double-ko reference in the point-by-point reponse for the previous revision (EMBOJ-2024-119820R_Point-by-point response).

*We agree the (ii) statement, 'it remains essential to demonstrate that MCM5 UFMylation (and associated fork phenotypes) are unaffected by UFBP1 or CDK5RAP3 depletion'. To address this point, we here depleted either UFBP1 or CDK5RAP3 in HeLa cells by using siRNAs. We found that neither siUFBP1 nor siCDK5RAP3 had detectable impact on the UFMylation of endogenous (**Graphic 1A and 1B**) or exogenous MCM5 (**Graphic 1C and 1D**), or the replication speed (**Graphic 1E and 1F**). These results suggested that the role of MCM5 UFMylation in replication fork progression do not need these ER-localized adaptor E3 components. Thus, the UFMylation pathway regulating replication fork dynamics operates via a*

distinct mechanism separable from ER-RQC-associated UFMylation. We appreciate your insight, which prompted this important validation.

Graphic 1

Referee #2:

This is a revised version of an earlier manuscript reporting MCM5 UFMylation and its effects on replication origin activation and fork progression, in which Z. Li et al have addressed my comments and criticisms. While most of the minor comments

have been properly solved, the new results presented in response to two of my major comments still require some clarification.

Response: Thank you for your careful review of our revised manuscript.

1. Analyses of inter-origin distance: the authors provide a better description of the methodology (stretched DNA fibers). From this updated description it is clear that no DNA antibody was used to control for fiber integrity. Instead, the authors relied on "fibers lying exactly on a horizontal line" to be considered part of the same DNA molecule. This approach is hardly rigorous for the reliable quantification of inter-origin distances. In the rebuttal letter, they do show an experiment that combined CldU, IdU and ssDNA stainings (the recommended approach) that reproduces the longer IODs upon siUFL1. However, the CldU staining in this case is spotty and hard to integrate with the IdU staining, making the identification of origin positions rather arbitrary. I realize that IOD experiments with stretched fibers are technically challenging (this was the point of my original criticism), but precisely for this reason the ssDNA antibody control is very important. One alternative way to back up the notion that fewer origins are activated upon siUFL1 without doing new experiments would be to quantify the relative number of red-green-red tracks relative to the total number of red-green forks tracks.

Response: Thank you very much for the reviewer's constructive: 'quantify the relative number or red-green-red tracks relative to the total number of red-green forks tracks.'. Accordingly, we have quantified the ratio of red-green-red (RGR) tracks to the total number of red-green (RG) fork tracks across biological replicates. This analysis, presented in the Graphic 2, demonstrates a statistically significant decrease in the RGR/RG ratio upon UFL1 knockdown (Revised Figure EV1H). This quantification also indicated the role of UFMylation in replication origin firing.

Revised Figure EV1H

2. In Figure 1G, I had asked to add the immunoblots with expression levels of the WT and the different mutants of UFM1, UFC1 and UBA5. They are shown as new panels

N-O-P in Figure S1. The results with UFM1 and UFC1 are clear, but two aspects of the results with UBA5 are confusing. First, the siRNA does not seem to downregulate endogenous UBA5 in most cases (assuming it is the lower band under 50 KD). Second, the levels of one of the mutants (Q302Ter*) are not shown, most likely because this is a truncation mutant with a different molecular weight - but in this case a larger section of the gel could be shown. These issues need to be discussed and clarified.

Response: Thank you for highlighting these first and second points.

First, we agree that 'the siRNA does not seem to downregulate endogenous UBA5 in most cases'. We have solved this problem by using an alternative UBA5 antibody (UBA5 polyclonal antibody, Proteintech, 12093-1-AP) (See Revised Figure EV1P).

Second, we failed to detect the Q302Ter mutant, presumably because the antibody recognizes the C-terminal region of UBA5, which is absent in the Q302Ter* truncation mutant. We then confirmed Q302Ter expression (~34 kDa) by utilizing an anti-MYC antibody targeting the MYC tag present on all transfected constructs at the N terminal (see the last lane below). These data are now included in the revised*

Figure EV1P.

Referee #3:

I think the revised manuscript is satisfactory for publication just by correcting a few minor points as follows.

Response: Thank you for the positive comment.

1. In Figure 6B, Inter-origin Distances (μM) must be Inter-origin Distances (μm).

Response: We apologize for this oversight and have corrected this clerical error in revised Figure 6B.

2. In Figure 7A, the layout of the experimental conditions in horizontal axis was not fitted with the graph.

Response: Thank you for spotting this error; we have adjusted the horizontal axis layout in Figure 7A to better fit the graph.

3. In Figure EV2, it might be better to add some explanation about UFBP1 and UFSP2 (and CDK5RAP3).

Response: We agree with this point and have updated the revised manuscript accordingly "we next explored the specific role of UFMylation in this process. First, we analyzed the association of UFMylation E3 ligase complex UFL1-UFBP1-CDK5RAP3 (Makhlouf et al. 2024a; DaRosa et al. 2024a), and the main de-UFMylase UFSP2 (Ishimura et al. 2017) with genomic DNA during the cell cycle".

4. In line 13 of page 18, "This reduction in the proliferation rate likely ---" should be "This reduction in the progression rate likely ---".

Response: Thank you for spotting this error; we have amended this phrase in our revised manuscript.

Dr. Xingzhi Xu
Shenzhen University Medical School
Cell Biology
1066 Xueyuan Blvd, Rm. A6-901
Shenzhen University Lihu Campus
Shenzhen, Guangdong 518055
China

27th Aug 2025

Re: EMBOJ-2024-119820R1
MCM5 UFMylation regulates replication origin firing and fork progression

Dear Drs. Xu and Takeda,

Thank you for submitting your final revised manuscript for our consideration. I am pleased to inform you that we have now accepted it for publication in The EMBO Journal.

With kind regards,

Hartmut
